# Bacterial actin MreB forms antiparallel double filaments

Fusinita van den Ent[1][*][†], Thierry Izoré[1][†], Tanmay AM Bharat[1], Christopher M Johnson[2], Jan Löwe[1]

[1]Structural Studies Division, Medical Research Council - Laboratory of Molecular Biology, Cambridge, United Kingdom; [2]Protein and Nucleic Acid Chemistry Division, Medical Research Council - Laboratory of Molecular Biology, Cambridge, United Kingdom

**Abstract** Filaments of all actin-like proteins known to date are assembled from pairs of protofilaments that are arranged in a parallel fashion, generating polarity. In this study, we show that the prokaryotic actin homologue MreB forms pairs of protofilaments that adopt an antiparallel arrangement in vitro and in vivo. We provide an atomic view of antiparallel protofilaments of *Caulobacter* MreB as apparent from crystal structures. We show that a protofilament doublet is essential for MreB's function in cell shape maintenance and demonstrate by in vivo site-specific cross-linking the antiparallel orientation of MreB protofilaments in *E. coli*. 3D cryo-EM shows that pairs of protofilaments of *Caulobacter* MreB tightly bind to membranes. Crystal structures of different nucleotide and polymerisation states of *Caulobacter* MreB reveal conserved conformational changes accompanying antiparallel filament formation. Finally, the antimicrobial agents A22/MP265 are shown to bind close to the bound nucleotide of MreB, presumably preventing nucleotide hydrolysis and destabilising double protofilaments.

**\*For correspondence:** fent@mrc-lmb.cam.ac.uk

[†]These authors contributed equally to this work

**Competing interests:** The authors declare that no competing interests exist.

**Reviewing editor**: Michael Laub, Massachusetts Institute of Technology, United States

## Introduction

Cell shape is a characteristic and hereditary feature that is crucial for existence and its regulation is a common challenge for organisms across all biological kingdoms. Shape in eukaryotic cells is provided by the cytoskeleton that consists of actin, tubulin, and intermediate filaments. Prokaryotes also have a dynamic, filamentous network of proteins, which are homologous to the eukaryotic cytoskeletal elements (*Löwe and Amos, 2009*). In non-spherical bacteria, the actin homologue MreB (*Jones et al., 2001*; *van den Ent et al., 2001*) is essential for shape maintenance as depletion of MreB through genetic knockouts or MreB-targeted drug treatment results in misshapen cells that eventually lyse (*Wachi et al., 1987*; *Levin et al., 1992*; *Lee et al., 2003*; *Bendezú and de Boer, 2008*). Incorrect localisation of MreB adversely affects polar targeting of protein complexes and, in some organisms, chromosome segregation (*Gitai et al., 2004*, *2005*; *Wagner et al., 2005*; *Cowles and Gitai, 2010*). Vital for MreB's role in cell shape determination is the interaction with the elongasome, the protein complex orchestrating peptidoglycan cell wall morphogenesis. The elongasome consists of MreC and MreD (MreB's operon partners), RodA, PBP1A, PBP2, RodZ, as well as MurF, MurG and MraY (*Alyahya et al., 2009*; *Favini-Stabile et al., 2013*; *Gaballah et al., 2011*; *Holtje, 1996*; *Mattei et al., 2010*; *Mohammadi et al., 2007*; *Shiomi et al., 2008*; *Szwedziak and Löwe, 2013*; *van den Ent et al., 2006*; *van den Ent et al., 2010*). The elongasome ensures that new glycan strands cross-linked by short peptides are inserted along the peptidoglycan scaffold during cell elongation (*den Blaauwen et al., 2008*; *Goffin and Ghuysen, 1998*; *Matsuhashi et al., 1990*; *Scheffers et al., 2004*; *Scheffers and Pinho, 2005*). The peptidoglycan layer provides structural integrity to the cell wall and counteracts turgor-induced lysis.

The existence of helical, cytomotive bundles of MreB originally observed along the length of rod-shaped cells (*Jones et al., 2001*; *Carballido-López and Errington, 2003*; *Figge et al., 2004*;

**eLife digest** Bacterial cells come in a variety of different shapes, including spheres, rods, spirals, and crescents. Shape is important for bacterial cells because it plays a role in cell division, helps to maximize the uptake of nutrients, and aids cell movement.

The shape of a cell is determined mainly by its cytoskeleton, a form of 'scaffolding' within the cell that is made of various protein filaments. The bacterial cytoskeleton was discovered over 20 years ago, but it has not been studied as much as the cytoskeletons of yeast, plant, animals, and other eukaryotes.

Many of the bacterial proteins and filaments that make up the cytoskeleton are similar to those found in eukaryotes. A protein called MreB, for example, is the bacterial equivalent of actin, which performs a wide range of roles in eukaryotes. However, van den Ent, Izoré et al. have now shown that the detailed structure of MreB filaments is different to that of actin filaments.

It has been known for some time that actin filaments are composed of two strands of actin proteins that are twisted and parallel with each other. MreB filaments are also made of two strands of MreB proteins, but van den Ent, Izoré et al. found that these strands are straight, not twisted, and that they are antiparallel rather than parallel. Thus, unlike other filaments of actin-like proteins, where the two ends of the filament are clearly different from each other, the antiparallel strands of MreB form a double filament without a clear direction.

van den Ent, Izoré et al. also showed that MreB double filaments can bind to surfaces that mimic a bacterial cell membrane, and that mutant bacterial cells without these double filaments adopt the wrong cell shape. Further experiments exposed potential targets on the MreB filaments for antibiotics that could treat bacterial infections.

*Gitai et al., 2004*; *Slovak et al., 2005*; *Vats and Rothfield, 2007*; *Löwe and Amos, 2009*) has been questioned by recent light microscopy studies that suggest MreB forms discontinuous fibres underneath the cell membrane that are driven by the elongasome itself (*Dominguez-Escobar et al., 2011*; *Garner et al., 2011*; *van Teeffelen et al., 2011*; *Olshausen et al., 2013*; *Reimold et al., 2013*). The MreB fibres are attached to the cell membrane via a hydrophobic loop in *Thermotoga maritima* as well as Gram⁺ bacteria and via an additional N-terminal amphipathic helix in Gram⁻ bacteria (*Salje et al., 2011*; *Maeda et al., 2012*). The tendency of purified MreB from Gram⁻ bacteria to aggregate can be attributed to the presence of these hydrophobic sequences. Hence, previous biochemical and structural work has focussed on MreB from *T. maritima* (TmMreB) that can be obtained in much higher quantities than *E. coli* MreB (*Nurse and Marians, 2013*) or *B. subtilis* MreB (*Mayer and Amann, 2009*). MreB assembles into straight, double protofilaments in the presence of ATP or GTP and these can gather laterally into sheets (*Nurse and Marians, 2013*; *Popp et al., 2010b*; *van den Ent et al., 2001*). Although the architecture of a single protofilament is surprisingly conserved within the actin family of proteins (*van den Ent et al., 2001*; *Gayathri et al., 2013*; *Popp et al., 2010c, 2012*; *Szwedziak et al., 2012*; *van den Ent et al., 2002*), the assembly of protofilaments into higher order structures differs greatly (*Ozyamak et al., 2013b*). The overall architecture has implications for filament polarity, which affects differential growth of the filament ends and directional movement. Force-generating motors that push plasmids apart all contain pairs of parallel helical protofilaments as their basic building block (*Gayathri et al., 2012, 2013*; *Polka et al., 2009*; *Popp et al., 2010a, 2010c, 2012*; *van den Ent et al., 2002*; *Galkin et al., 2012*). Presumably, the membrane-binding ability of MreB and FtsA constrains their filament architecture to straight (or slightly bent), rather than helical, pairs of protofilaments. But how are these filaments arranged?

Here, we demonstrate that MreB from *Caulobacter crescentus* forms pairs of antiparallel protofilaments as revealed by crystal structures, an architecture unprecedented among actin-like proteins. The antiparallel arrangement ensures that the amphipathic helix of each protofilament binds to membranes as shown by cryo-electron microscopy (cryo-EM). Pairs of protofilaments are essential for cell shape maintenance in *Escherichia coli* and these are shown to be oriented in an antiparallel fashion by in vivo cross-linking. The mechanism for protofilament formation of *C. crescentus* MreB is elucidated by crystallography and shown to be reminiscent of that of other actin-like proteins, revealing the typical domain closure upon polymerisation. Finally, co-crystallisation studies of *Caulobacter* MreB with

the antimicrobial reagents A22 and MP265 explain their inhibitory mechanism and shed light on the activation mechanism of ATP hydrolysis during polymerisation.

## Results

### Crystal structure of *Caulobacter* MreB reveals double, antiparallel filaments

Structural insight into MreB from mesophilic bacteria has been hampered by the tendency of the proteins to aggregate upon purification, but through biochemical and bioengineering tools sufficient quantities of stable versions of *Caulobacter crescentus* MreB were obtained (*Table 1*) that were amenable to structural studies. Deleting the N-terminal amphipathic helix and mutating the residues in the hydrophobic loop (*Figure 1—figure supplement 1*) make the protein (ΔCcMreBdh) less prone to aggregation. The crystal structure of *Caulobacter* MreB (ΔCcMreBdh) reveals the typical actin fold, containing two domains on either side of a nucleotide binding cleft (*Figure 1A*; *Kabsch and Holmes, 1995*). Both domains are separated into two subdomains, A and B. Domain II is structurally the most preserved domain within the actin family of proteins, whereas domain IB is the most diverse. But even this domain is very similar in both *Caulobacter* MreB and *Thermotoga maritima* MreB (TmMreBh, *van den Ent et al., 2001*), superimposing with an RMSD of 0.97 Å (*Figure 1A*). The protofilament architecture is also very well conserved, with an identical subunit repeat distance of 51.1 Å and a similar arrangement of the longitudinal interface (*Figure 1C*). However, only single protofilaments were present in the *Thermotoga* MreB crystals, despite the fact that electron microscopy images clearly show doublets (*van den Ent et al., 2001*). In contrast, the crystals of *Caulobacter* MreB contain doublets that, remarkably, crystallised in an antiparallel arrangement (*Figure 1D*). The electrostatic surface potential of the double protofilaments reveals the hydrophobic loop and the basic charges surrounding the membrane-binding site (*Figure 1B*).

### *Caulobacter* MreB binds as pairs of protofilaments to lipids

Full-length, untagged MreB from *C. crescentus* (CcMreB) binds to a lipid monolayer as a double proto-filament as shown by negative stain electron microscopy (EM, *Figure 2A*), similarly to TmMreB (*Salje et al., 2011*). 2D averaging of images of negatively stained filaments shows double protofilaments interacting with their flat sides (*Figure 2B*), which coincides with other, double-helical actin-like protein filaments (*Popp et al., 2010a, 2010c, 2012, 2012*; *Salje et al., 2011*; *Gayathri et al., 2013*). Although the resolution is low, the entire double protofilament structure of ΔCcMreBdh fits nicely into the 2D averaged EM images with a longitudinal subunit repeat of 5.1 nm and the widths of the doublet encompassing 6.5 nm (*Figure 2C*). The previously reported placement of two single protofilaments of TmMreB into 2D averaged filaments, though merely an educated guess at that time (*Salje et al., 2011*), turned out to be right. By introducing a single point mutation in the interprotofilament interface of CcMreB (V118E), single protofilaments are formed as shown by negative stain EM (*Figure 2D*). Pairs of protofilaments of CcMreB distort liposomes as seen by cryo-EM (*Figure 2E*; *Video 1*) and coat the membrane extensively (compare *Figure 2G* to uncoated membranes in *Figure 2F*). The irregular arrangement of CcMreB filaments on liposomes makes it unsuitable for subtomogram averaging. Hence, we turned to rigid lipid tubes made of *E. coli* total lipid extract mixed with C24:1 β-D galactosyl ceramide (*Parmenter and Stoilova-McPhie, 2008*; *Parmenter et al., 2008*). These tubes remain straight in the presence of MreB and are suitable for obtaining higher resolution reconstructions revealing a side view that shows how the protein coats the membrane (*Figure 3A–D*; *Video 2*). Although overall appearances are very similar, TmMreB forms much straighter protofilaments on lipid tubes than CcMreB (*Figure 3C*), making TmMreB more amenable to subtomogram averaging (*Figure 3E–G*). Both 2D and 3D averaging unambiguously show how close an MreB

**Table 1.** Summary of proteins used in this study

| | |
|---|---|
| *Caulobacter crescentus* MreB (GenBank: ACL95077.1) | |
| CcMreB | M1-CcMreB-A347 |
| ΔCcMreBh | M-I9-CcMreB-A347-GSHHHHHH |
| ΔCcMreBdh | M-I9-CcMreB(F102S, V103G)-A347-GSHHHHHH |
| ΔCcMreBdmh | M-I9-CcMreB(F102S, V103G, S283D)-A347-GSHHHHHH |
| *Thermotoga maritima* MreB1 (GenBank:AAD35673.1) | |
| TmMreB | M1-TmMreB-G336 |
| TmMreBh | M1-TmMreB-G336-GSHHHHHH |

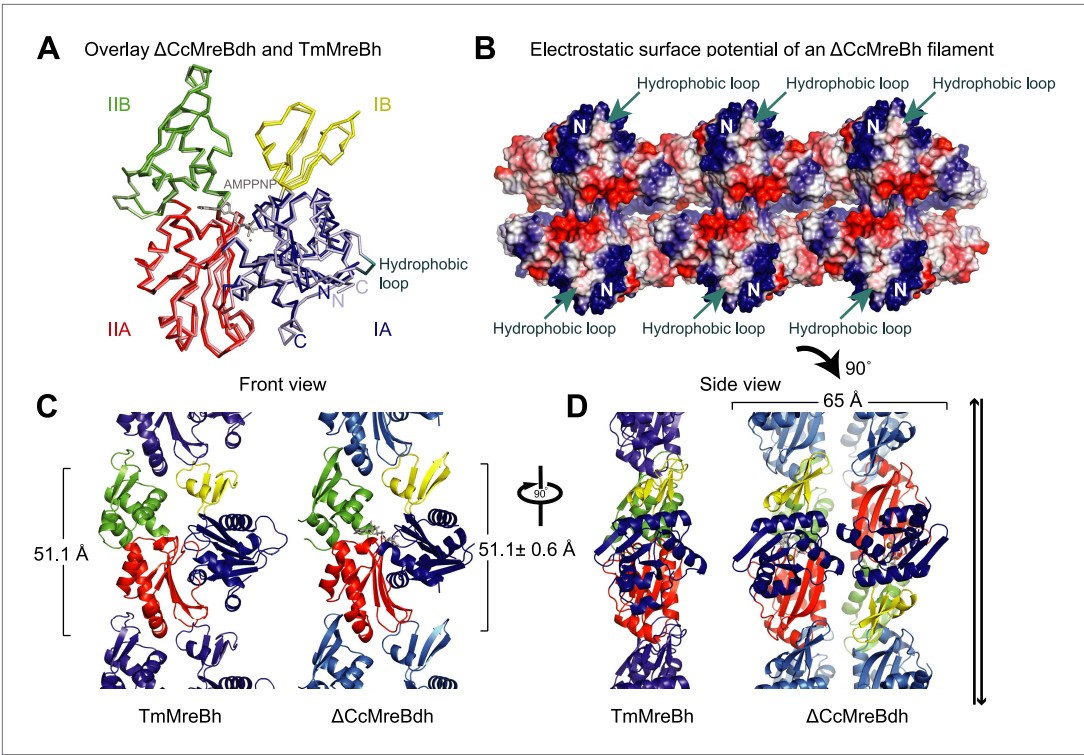

**Figure 1**. Comparison between crystal structures from *Caulobacter crescentus* and *T. maritima* MreB. (**A**) The crystal structure of *C. crescentus* MreB (ΔCcMreBdh, d_3 [see **Table 2**] shown in dark colours) superimposes on that of *T. maritima* MreBh (PDB code 1JCG, shown in pastel colours) with an overall RMSD of 0.97 Å over 310 aligned residues and a Z-value of 20.95. The sequence similarity between the two proteins is 55.5%. Domains are coloured blue, for subdomain IA, yellow for subdomain IB, red for subdomain IIA, and green for subdomain IIB. The active site is located in the interdomain cleft and has AMPPNP (shown in grey) and $Mg^{2+}$ (pink) bound to ΔCcMreBdh. The hydrophobic loop (coloured in cyan) is shown for TmMreB and is not ordered in ΔCcMreBdh (d_3). (**B**) Electrostatic surface potential of an antiparallel protofilament of *C. crescentus* MreB shown from side view, where it binds to the membrane. The hydrophobic loop is indicated by an arrow (cyan) and the N-terminus from where the amphipathic helix emerges is shown (N). Positively charged regions are shown in blue, negatively charged areas in red and hydrophobic patches in white. Crystals containing antiparallel protofilaments of ΔCcMreBh were soaked in ATPγS. (**C**) Front view of a protofilament from *T. maritima* MreB and *C. crescentus* MreB. Crystals containing TmMreBh protofilaments (PDB code 1JCG, left) have the same subunit repeat as those from ΔCcMreBdh (right, d_3). The colour code is as in *Figure 1A*. The standard deviation is calculated based on 7 crystal structures containing protofilaments of *C. crescentus* MreB (**Table 2**). (**D**) Side view of a single protofilament of *T. maritima* MreB (PDB code IJCG, left) and double protofilaments from ΔCcMreBdh (right, d_3) arranged in an antiparallel fashion (indicated by the black arrows on the right). Colour codes indicate the four subdomains, as explained in *Figure 1A*.

The following figure supplements are available for figure 1:

**Figure supplement 1**. Sequence alignment between MreB from *C. crescentus* (CCM), *E. coli* (ECM), and *T. maritima* (TMB).

**Figure supplement 2**. Schematic showing lack of polarity at filament ends in an MreB filament.

filament sits on the tubes, extending the thickness of the density including the lipid bilayer to 12.5 nm (*Figure 3E*). As shown in *Figure 3E* and *Figure 3G*, the bilayer is resolved well and the longitudinal repeat distance of the columns protruding from the outer leaflet is in agreement with the expected subunit repeat of 5.1 nm in TmMreB protofilaments. Also, subtomogram averaging clearly demonstrates that membrane binding of MreB distorts the outer, but not the inner leaflet of the lipid bilayer, which is in agreement with the proposed binding mode of TmMreB via a hydrophobic loop. It is not very surprising to us that MreB has not been seen in cells by cryo-EM (*Swulius et al., 2011*; *Swulius and Jensen, 2012*), as tomograms of cells are usually taken at rather low magnification and high

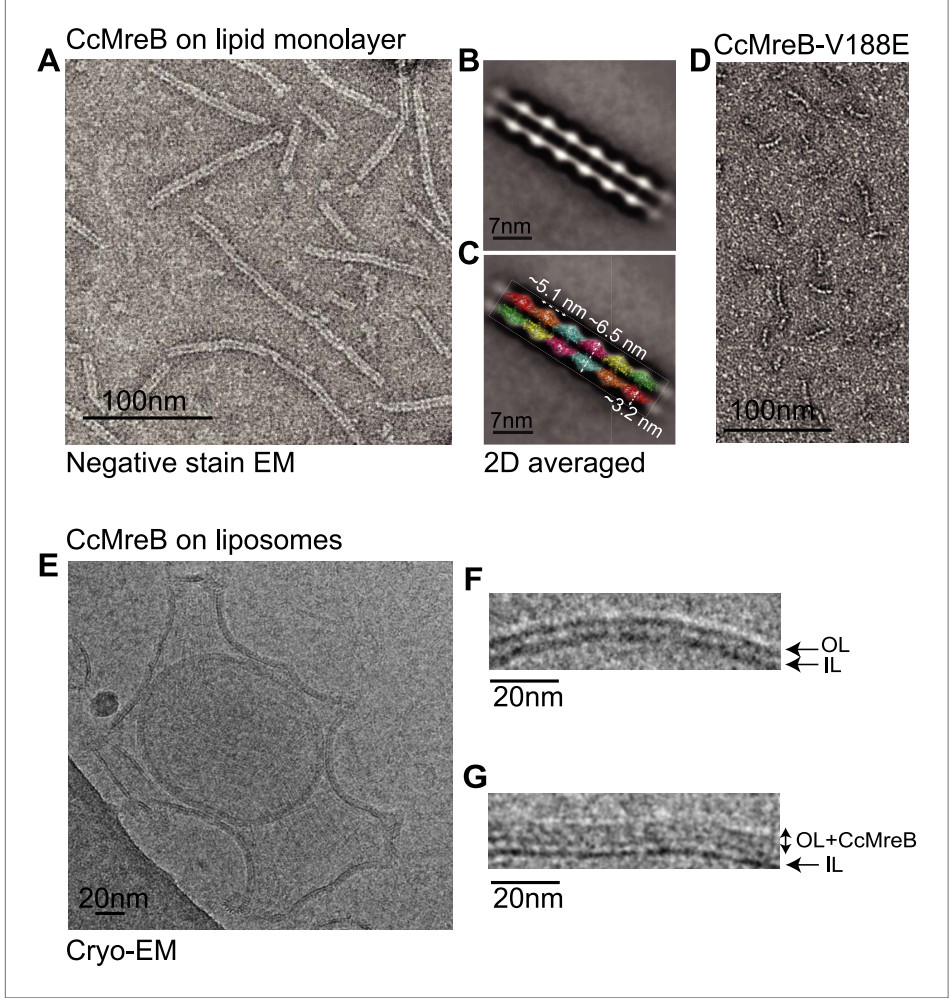

**Figure 2**. MreB forms double protofilaments on a lipid bilayer and distorts liposomes. (**A**) Negative stained electron microscopy image of untagged CcMreB double protofilaments assembled on lipid monolayer at 0.25 mg/ml in the presence of ATP. Bar: 100 nm. (**B**) 2D averaging of CcMreB double protofilaments from views similar to the one shown in **A**. Bar: 7 nm. (**C**) Scaled double protofilament from the ΔCcMreBh crystal structure (d_e) was positioned into the EM density obtained from the 2D averaging. Bar: 7 nm. (**D**) Negative stained electron microscopy image of untagged CcMreB-V118E single protofilaments assembled on lipid monolayer at 0.13 mg/ml in the presence of ATP. Bar: 100 nm. (**E**) Cryo-EM image of ATP-bound CcMreB (1 mg/ml), which distorts liposomes made from *E. coli* total lipid extract. Scale bar: 20 nm. (**F**) Negative control showing the membrane of a liposome in the absence of CcMreB. OL: outer leaflet. IL: inner leaflet. Scale bar: 20 nm. (**G**) View of a liposome membrane in the presence of ATP-bound CcMreB. OL + CcMreB: outer leaflet with CcMreB. IL: inner leaflet. Scale bar: 20 nm.

defocus, leading to a resolution lower than what is needed to resolve the leaflets of the lipid bilayer, let alone, an MreB filament from the protein-embedded membrane.

## Genetic analysis of mutations that disrupt longitudinal and lateral contacts within the double filament of MreB

A first indication that a doublet of protofilaments is required for MreB's function comes from genetic studies in *E. coli* (*Figure 4*). An MreBCD knockout strain (FB17), otherwise not viable, survives if complemented with extra FtsZ and a copy of a plasmid expressing MreC and MreD (pλ::*mreCD*; pFB124) (*Bendezú and de Boer, 2008*; *Salje et al., 2011*). Additional FtsZ is provided indirectly from a second plasmid (pFB112) that constitutively expresses the transcription factor SdiA, ensuring elevated levels of FtsQAZ (*Bendezú and de Boer, 2008*). Cells become rod-shaped and lose their dependency on elevated levels of FtsZ if complemented with a third plasmid carrying a functional copy of *mreB* under the *lac* promoter

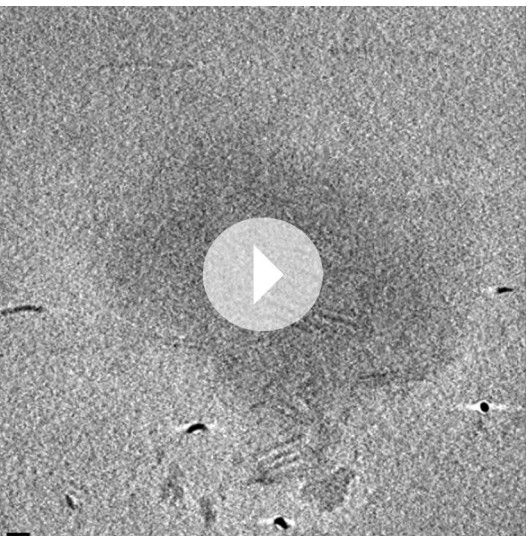

**Video 1**. CcMreB distorts vesicles. Tomography of full-length untagged MreB (CcMreB) filaments (+ATP) bound to vesicles made from *E. coli* total lipid extract. MreB filaments are clearly seen on both the top and bottom flattened surfaces of the vesicle. Scale bar: 20 nm.

(pFB209) (*Figure 4A*). This complex genetic set up enables investigation of the phenotype of MreB containing mutations that might render the protein non-functional, since MreB is essential in *E. coli*. Altered versions of MreB were introduced that carried a mutation disrupting either the intra- or interprotofilament interface (EcMreB-S284D or EcMreB-V121E, respectively, *Figure 4D*), as deduced from the crystal structure. Indeed, a single mutation in the intra-protofilamant interface renders the protein monomeric as shown by structural analysis of the equivalent mutant S283D in CcMreB (presented below), whereas a single mutation in the interprotofilament interface results in single protofilaments as discussed above (CcMreB-V118E, *Figure 2D*). In the absence of MreB, cells are spherical (first column *Figure 4B*), but they revert back to rod shape when expressing wild-type MreB (second column *Figure 4B*). When either the intra- (S284D) or inter-protofilament (V121E) interface is disrupted, cells remain spherical (third and fourth column *Figure 4B*, *Figure 4C*). These results demonstrate firstly that MreB protofilament formation is essential for cell shape maintenance and secondly that these protofilaments need to interact with their flat sides to make functional MreB filaments.

## In vivo cross-linking shows antiparallel arrangement of *E. coli* MreB filaments

Thus far we have established that pairs of protofilaments are required for functional MreB in vivo and that these protofilaments interact with their flat sides (*Figure 4*). To verify that the antiparallel arrangement of protofilaments observed both in EM and in the crystal structure reflects the functional state of MreB in cells, site-specific cross-linking was performed. Cysteine residues were introduced at specific sites in and outside the protofilament interface and mutated MreB was expressed in the MreB knockout strain as described above. But before that, cysteine residues present in native MreB had to be mutated to serine residues to ensure that any cross-linked product would arise from the newly introduced cysteine(s). As shown in *Figure 5—figure supplement 1*, the cysteine-free form of MreB fully complemented an *mreB* knockout strain. At late log phase, the cysteine mutants were subjected to the thiol-reactive cross-linker bismaleimidoethane (BMOE). Thiol-reactive cross-linkers work by forming covalent bonds between two cysteine residues. As the protofilaments in the antiparallel arrangement are symmetry-related, certain single cysteine locations may produce a cross-linked dimer as long as the distance between the mutated residues is within reach of the cross-linker arm (~8 Å). As shown in *Figure 5*, cross-linked dimers were obtained when cysteine residues were introduced in the protofilament interface (*Figure 5B*, lanes 1–7, 9–10, 12, 13, 14), but not when cysteine residues were located outside this interface (*Figure 5B*, lanes 16–19), nor in the absence of any cysteine (*Figure 5B*, lane 15). When a pair of cysteine residues was introduced on opposite sides of the antiparallel protofilament interface, stronger bands for the cross-linked dimers were repeatedly obtained than those of the sum of each mutant alone, providing further proof of the antiparallel arrangement (*Figure 5B*, lanes 8 and 11, compared to lanes 6–7 and 9–10, respectively).

The phenotypes of the mutants were analysed at the same time as the cross-linking was performed—at late log phase—a time point previously shown to give good complementation with wild-type MreB (*Bendezú and de Boer, 2008*; *Salje et al., 2011*). The cysteine-free mutant of MreB shows good complementation, as do some, but not all of the cysteine mutants (*Figure 5—figure supplement 1*). The lack of complementation of some of the mutants does not correlate with their ability to form filaments, indicating that analysis of the phenotype, which is known to be exquisitely sensitive, does probe more features than can be attributed to MreB polymerisation alone, such as filament dynamics and the interaction with host proteins.

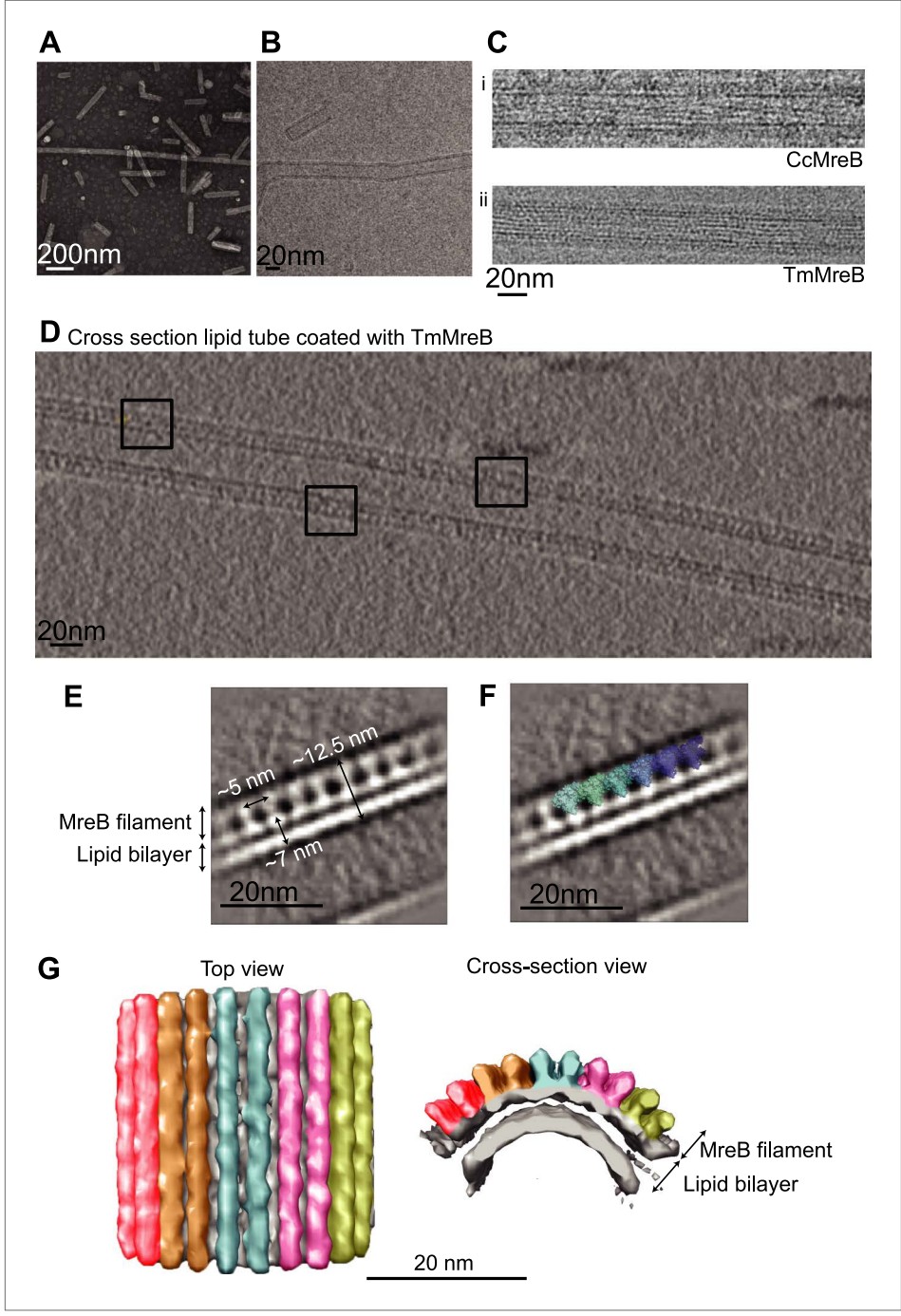

**Figure 3**. MreB binds to one leaflet of the lipid bilayer in the presence of ATP. (**A**) Image of negatively stained lipid tubes, made by mixing *E. coli* total lipid extract with 30% of D-Galactosyl-β1-1'-N-Nervonoyl-D-erythro-sphingosine (C24:1 β-D Galactosyl-Ceramide, Avanti Polar Lipids). Bar: 200 nm. (**B**) Cryo-EM image of a lipid tube. Bar: 20 nm. (**C**) Cryo-EM image of CcMreB (i), TmMreB (ii) on lipid tube at 1 mg/ml. Bar: 20 nm. (**D**) Cross section view of a lipid tube coated by TmMreB (1 mg/ml). Bar: 20 nm. (**E**) 2D averaging of the edge of TmMreB-coated lipid tubes from views such as those squared in *Figure 3D*. The longitudinal subunit repeats of ~5 nm is in agreement with that of the crystal structure (*Figure 1C*). The lipid bilayer is ~7 nm and MreB extends the density to a total of ~12.5 nm. Bar: 20 nm. (**F**) Scaled protofilament from the ΔCcMreBh crystal structure (d_3) was positioned into the EM density obtained from 2D averaging. Scale bar: 20 nm. (**G**) Subtomogram averaging reconstruction of a lipid tube coated with TmMreB. Top and cross-section views. Scale bar: 20 nm.

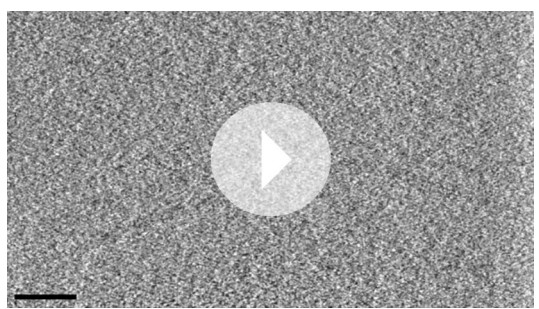

**Video 2**. TmMreB filaments on lipid nanotubes. Tomography of full length untagged MreB (TmMreB) filaments (+ATP) bound to lipid nanotubes. Filaments are straight and run parallel to the long axis of the tubes. Scale bar: 50 nm.

## What is MreB's mechanism for polymerisation into filaments?

To form a detailed picture of the requirements of MreB to polymerise, the full polymerisation cycle of MreB was investigated by X-ray crystallography. Crystal structures of CcMreB in different nucleotide states, either as a monomer or as a polymer, were determined to a resolution better than 2 Å (*Figure 6A*). Most crystal structures were obtained using ΔCcMreBdh, which behaved biochemically better than the full-length protein, yet was still showing correct filament formation as presented in *Figure 6—figure supplement 1*. A single mutation in the intra-protofilament interface (S283D, ΔCcMreBdmh) renders the protein monomeric, as shown both by crystallography (*Figure 6*) and by EM (*Figure 6—figure supplement 1*). This mutant turned out to be most useful in studying the polymerisation cycle. From looking at the crystal structures of MreB in different nucleotide states, it immediately becomes apparent that surprisingly small structural changes can trigger MreB polymerisation. Monomeric CcMreB shows little structural movement between the ADP- and the ATP analogue AMPPNP-bound states (*Figure 6D*) with domains I and II overlapping with an RMSD of 1.58 and 0.57 Å, respectively (*Figure 6B*). An initial domain closure is observed upon binding of AMMPNP (*Figure 6D*) by the monomer. This movement progresses into a full propeller twist once it reaches the double protofilament state (*Figure 6E*). Upon polymerisation, domain IB shows a rotation of 21.7° towards the nucleotide binding cleft as analysed by Dyndom (*Figure 6—figure supplement 2A*; *Hayward and Lee, 2002*). This domain closure is accompanied by a twist between domains IA and IIA, as can be appreciated from the top view in *Figure 6—figure supplement 2D* and from the schematic in *Figure 6C*. The relatively small conformational change upon polymerisation has a dramatic effect on the chemistry of the active site. In the AMPPNP protofilament conformation, the active site residues E140 and T167 coordinate a water molecule that is in line for the hydrophilic attack on the gamma phosphate (*Figure 6F*). In the monomeric state, the active site residue E140 of MreB has moved away from the nucleotide by just 1 Å. But consequently, it can no longer coordinate the attacking water, which is absent in this structure, and thus renders the monomer inactive (*Figure 6F*). Once MreB is in the polymeric state, the presence or absence of nucleotide has very little effect on the overall structure, which is presumably stabilised by inter- and intra-protofilament contacts. The mechanism of disintegration of the polymer remains to be elucidated.

## Inhibitors A22 and MP265 block phosphate release in *Caulobacter* MreB

The MreB inhibitor S-(3, 4-dichlorobenzyl) isothiourea (A22) and its less cytotoxic and much more water-soluble derivative MP265 have been used extensively in cell shape studies as they perturb cell morphology reminiscent of an MreB knock-out (*Iwai et al., 2002*; *Gitai et al., 2005*; *Bean et al., 2009*; *Takacs et al., 2010*). Analyses of revertants that are resistant to A22 suggested that the inhibitor targets the active site of MreB (*Gitai et al., 2005*; *Dye et al., 2011*). Here, we report the mechanism of the inhibitory action using biochemical and structural methods.

Isothermal titration calorimetry experiments were used to acquire binding constants for the inhibitor MP265 and nucleotides that bind to the active site of ΔCcMreBdh. It revealed that ΔCcMreBdh has the highest affinity for ATP ($K_d$ ~1 µM), followed by ADP ($K_d$ ~4 µM) and ATPγS ($K_d$ ~4.6 µM) and finally by AMPPNP ($K_d$ ~65 µM). The inhibitor MP265 binds to ΔCcMreBdh with a $K_d$ ~27 µM and, somewhat surprisingly, its binding affinity increases 20- to 30-fold in the presence of di- and trinucleotide phosphates (*Figure 7A–C*). This is in contrast to results published earlier, which reported that binding of A22 to TmMreB seems to be sterically incompatible with simultaneous binding of ATP, as was determined by modelling (*Bean et al., 2009*).

Co-crystallisation of A22 and ΔCcMreBdh shows unambiguously that the inhibitor binds in the presence of ADP (*Figure 7D*). The anomalous difference map reveals the positions of the chlorine atoms of A22 and the di-phosphate of ADP (*Figure 7D*). At 1.5 Å resolution, it absolutely defines the exact position of a small molecule like A22. Because MP265 is much more manageable, co-crystals of MP265

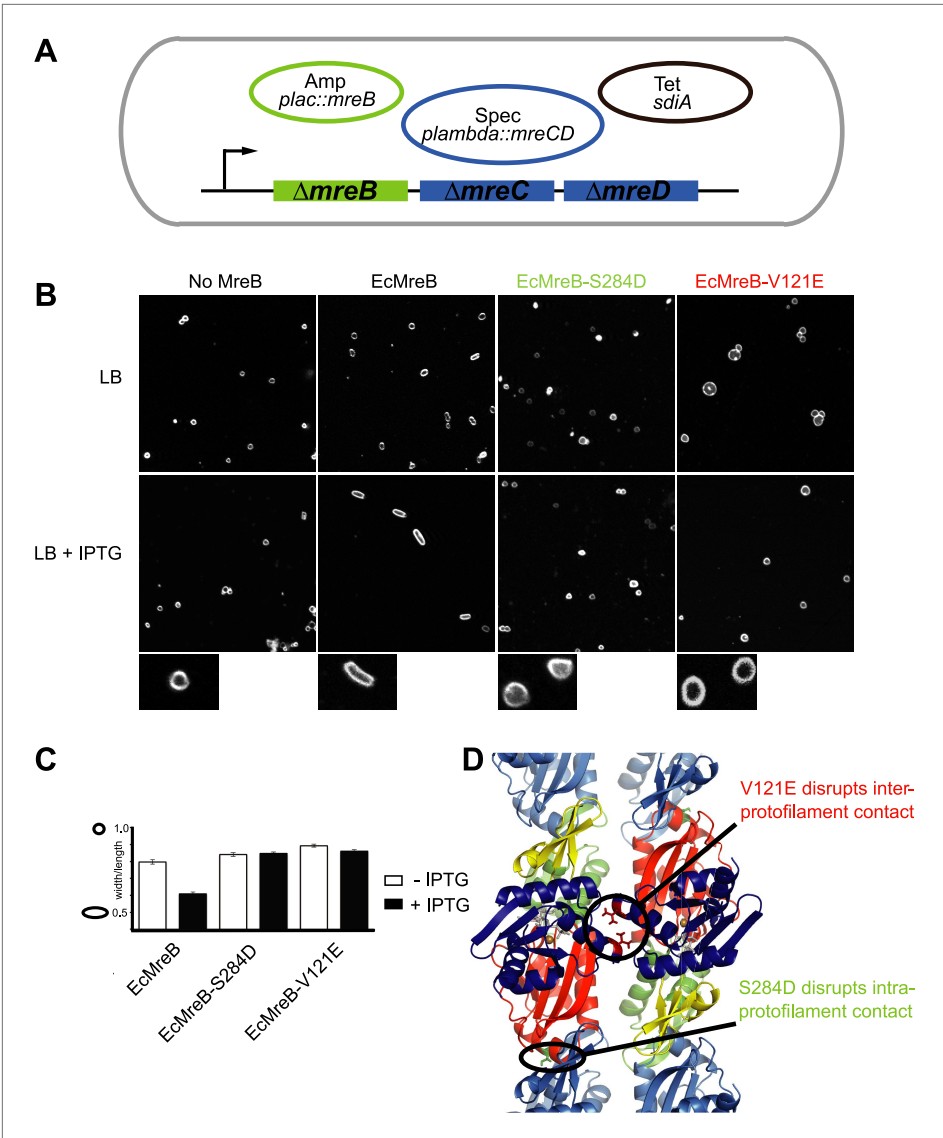

**Figure 4.** Pairs of MreB protofilaments are essential for cell shape maintenance in *E. coli*. (**A**) Schematic diagram showing the genetic set-up to examine cell shape maintenance by intra- and inter-protofilament mutants of MreB. An mreBCD knock-out strain (*mreBCD < > frt*, FB17) is kept alive by the constitutive expression of transcription factor SdiA (pFB112, tet[R], shown in black), that enhances levels of FtsQAZ and a plasmid carrying *mreCD* downstream of a lambda promoter and a temperature-sensitive repressor (pFB124, spec[R], blue). Variants of *mreB* are expressed from a lac promoter on a third plasmid (*plac::mreB*, amp[R], green). (**B**) Confocal images of strain FB17/pFB112/pFB124 (ΔmreBCD/tet sdiA/aadA clts *plambda:mreCD*) are shown in the first column, in the absence (top) and presence (bottom) of 250 μM IPTG. Wild-type MreB expressed from plasmid pFB209 (*plac::EcMreB*) complements the strain FB17/pFB112/pFB124 (ΔmreBCD/tet sdiA/aadA clts *plambda:mreCD*) in the presence (second column, bottom), but not in the absence (second column, top) of 250 μM IPTG. No complementation was observed with MreB that contains a single point mutation in the intra-protofilament interface (*plac::EcMreB-S284D*, pFE535) or in the inter-protofilament interface (*plac::EcMreB-V121E*, pFE400), regardless of the presence or absence of IPTG (third and fourth column). Cells were grown to late log phase and stained with FM4-64 prior to visualisation with a Zeiss confocal laser scanning microscope LSM510. (**C**) Cell shape distribution shown for strain FB17/pFB112/pFB124 (ΔmreBCD/tet sdiA/aadA clts plambda:mreCD) complemented by wild-type MreB (*plac::EcMreB*, pFB209, labelled EcMreB), by intra-protofilament mutant (*plac::EcMreB-S284D*, pFE535, labelled EcMreB-S284D) or by the inter-protofilament mutant (*plac::EcMreB-V121E*, pFE400, labelled EcMreB-V121E). The width/length ratio was determined computationally with ImageJ. Perfect round cells have a value of 1.0 and rod cells a value around 0.6 for *E. coli*. Total number of cells measured for each experiment are as follows: *plac::EcMreB* (FB17/pFB112/pFB124), n = 68

*Figure 4. Continued on next page*

*Figure 4. Continued*

(−IPTG); n = 92 (+IPTG); *plac::EcMreB-S284D* (FB17/pFB112/pFB124), n = 96 (−IPTG), n = 95 (+IPTG); *plac::EcMreB-V121E* (FB17/pFB112/pFB124), n = 51 (−IPTG), n = 51 (+IPTG). Error bars represent the standard error of the mean. (**D**) Structure of antiparallel protofilaments (d_3) showing the position of the intra-protofilament mutation S284D (green) and inter-protofilament mutation V121E (red).

and ΔCcMreBdh were obtained in the presence of both ADP and AMPPNP (**Figure 6A,B**, **Figure 7E,F**). The crystal structures of the complexes show that the inhibitor most likely prevents phosphate release in the MreB protofilament: MP265 blocks the exit channel for the phosphate and binds directly to the active site residue E140, preventing it from coordinating the catalytic water. In addition, it binds to the gamma phosphate (**Figure 7E**). Interestingly, crystals containing the MreB-inhibitor complex always contain single, rather than double protofilaments (**Figure 6B**; **Table 2**). A closer look at the inter-protofilament interface explains why double protofilaments can no longer form: as the inhibitor prevents the main dimerisation helix (Ala117-Ala130) from reaching into the interface, it causes

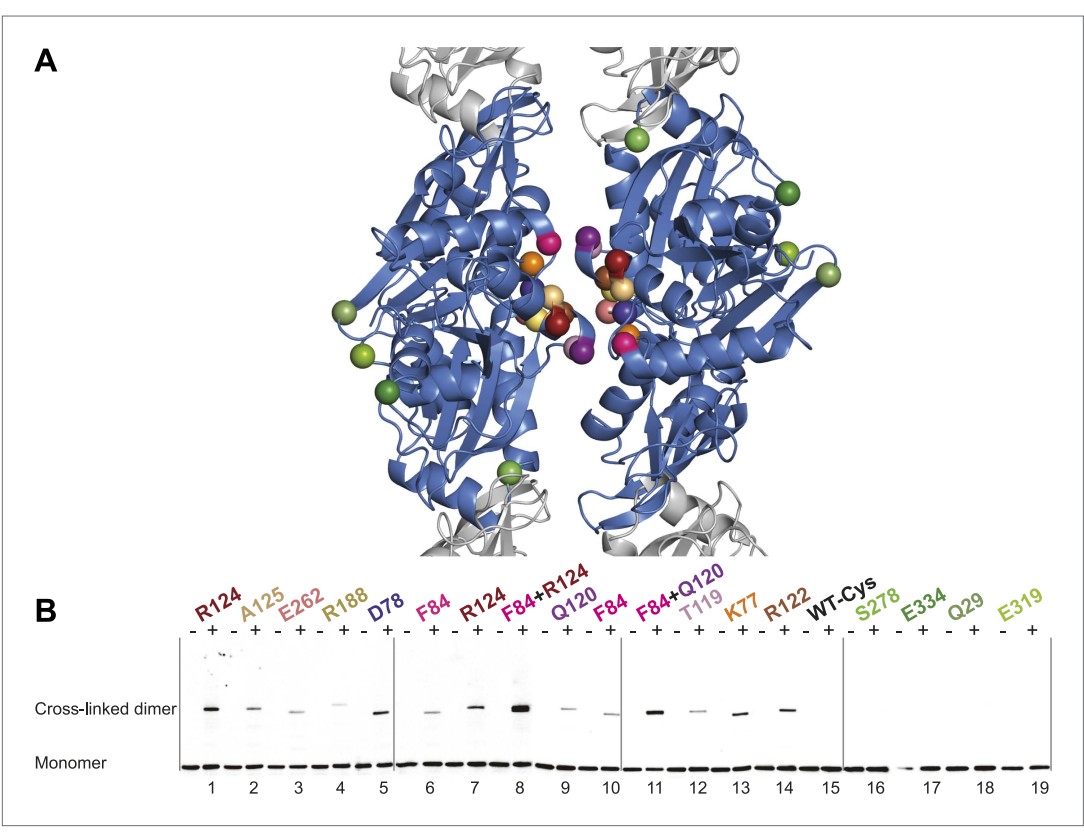

**Figure 5**. In vivo cysteine cross-linking of MreB in *E. coli*. (**A**) Antiparallel protofilament of ΔCcMreBh (d_e) as depicted as a ribbon plot, showing the protofilament interface mutations in bold coloured spheres and the mutations outside the interface in green coloured spheres. (**B**) Immunoblot after in vivo cross-linking using MreB-specific antibody shows samples from MreB knockout strains (FB17/pFB112/pFB124 (ΔmreBCD/tet sdiA/aadA clts plambda:mreCD)) supplemented with a cysteine-containing MreB allele (*plac::mreB*), containing mutation(s) as indicated above the lanes. Cells were grown to late log phase, then incubated on ice with (+) or without (−) the thiol-specific compound bismaleimidoethane (BMOE), prior to western blot analysis. A band corresponding to a dimer of MreB occurs with mutants that contain a cysteine in the dimer interface (indicated in bright colours). No cross-linking product was obtained in mutants that do not have cysteine or contain cysteine residues outside the dimer interface (green).

The following figure supplements are available for figure 5:

**Figure supplement 1**. Cell shape distribution.

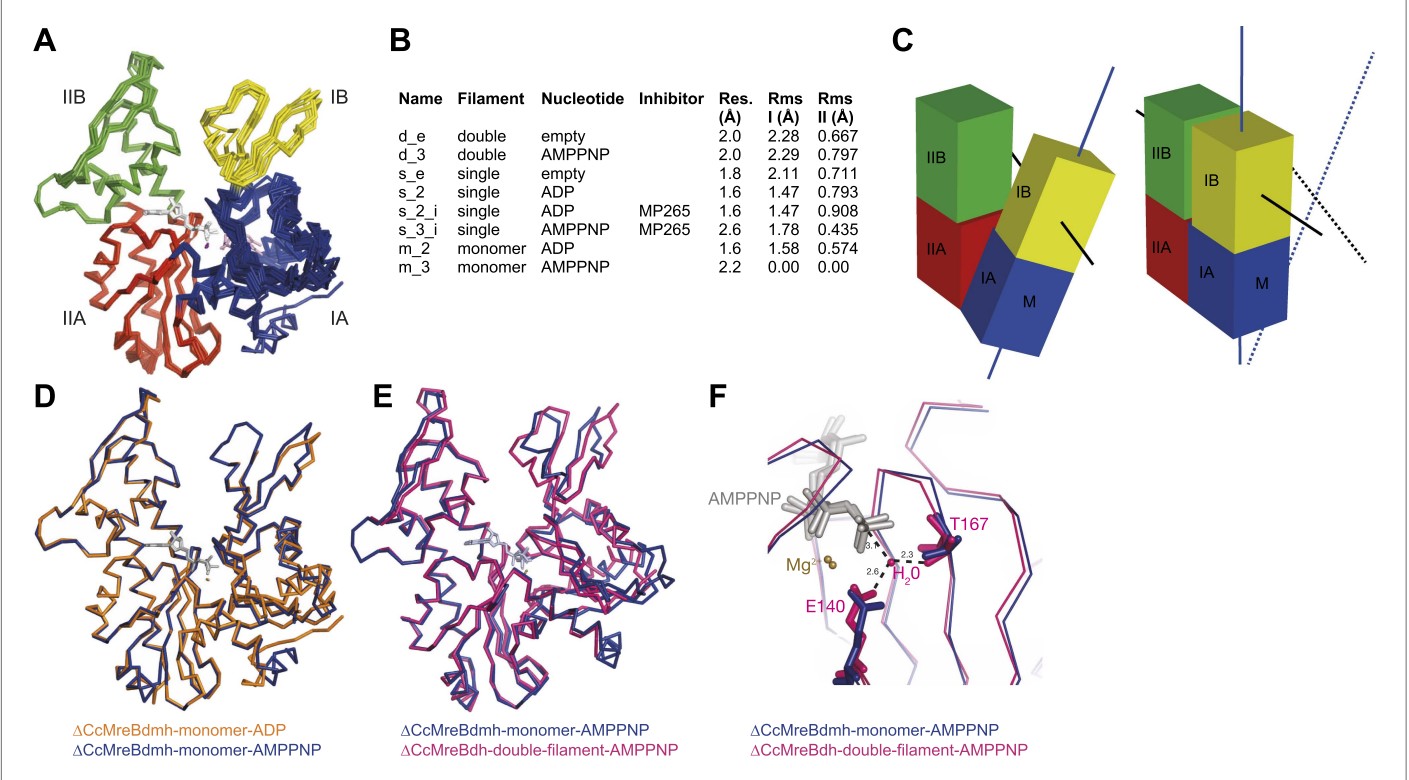

**Figure 6**. Crystal structures of MreB in different nucleotide states reveal a propeller twist. (**A**) Superposition of crystal structures of *C. crescentus* MreB in different nucleotide states as listed in *Figure 6B* show movement in domain I rather than domain II. Colour codes of subdomains are as in *Figure 1A*. The nucleotide is shown in light grey, $Mg^{2+}$ in purple, and the inhibitor in pink. (**B**) List of CcMreB crystal structures that occur either as a double or a single filament or as a monomer (second column), in the presence or absence of nucleotide/inhibitor (third and fourth column, respectively). The resolution of the structures is shown in the fifth column and the root mean square deviation (RMS) for domain I and domain II are listed in the last two columns. The RMS was calculated relative to the monomeric, AMPPNP-bound form (m_3 in *Table 2*). (**C**) Schematic drawing showing the propeller twist in MreB. The interdomain cleft narrows due to the movement of domain I towards domain II that is accompanied by a rotation of domain I resulting in flattening of the interfilament interface. Domain colours are described in *Figure 6A*, the membrane binding site is indicated (M). (**D**) Superposition of ADP-bound ΔCcMreBdmh (m_2, shown in orange) and AMPPNP-bound ΔCcMreBdmh (m_3, shown in blue). A small movement of domain IB initiates the propeller twist observed upon polymerisation (*Figure 6B*). (**E**) Superposition of AMPPNP-bound monomeric ΔCcMreBdmh (m_3, shown in blue) and AMPPNP-bound polymeric ΔCcMreBdh (d_3, shown in pink) reveals a propeller twist: closing of the nucleotide-binding cleft by the movement of subdomain IB towards subdomain IIB, accompanied by a twist of subdomain IA, resulting of flattening of the molecule (*Figure 6—figure supplement 2* and *Figure 1—figure supplement 2*). (**F**) Zoom into the active site of the superposition of AMPPNP-bound monomeric ΔCcMreBdmh (m_3, shown in blue) and AMPPNP-bound polymeric ΔCcMreBdh (d_3, shown in pink). The catalytic site residues E140 and T167 coordinate the attacking water to be in line with the gamma phosphate in filamentous MreB (pink). In monomeric, AMPPNP-bound MreB, the catalytic E140 moves 1 Å away from the nucleotide and is no longer able to bind to the catalytic water (blue).

The following figure supplements are available for figure 6:

**Figure supplement 1**. ΔCcMreBdh forms filaments as shown here by cryo-EM and a single mutation in the intra-protofilament interface disrupts filament formation (ΔCcMreBmdh).

**Figure supplement 2**. Domain rotations of actin-like proteins MreB, ParM and actin upon adopting the filament conformation.

weakening of the inter-protofilament contacts and likely reduces stability of the double protofilament (*Figure 7F*).

## Discussion

The antiparallel arrangement of protofilaments of *C. crescentus* MreB as revealed by the crystal structure (*Figure 1D*) and by EM (*Figure 2A–C*) is unprecedented within the actin family of proteins. It ensures that the amphipathic helix of both protofilaments can bind to the membrane and that both

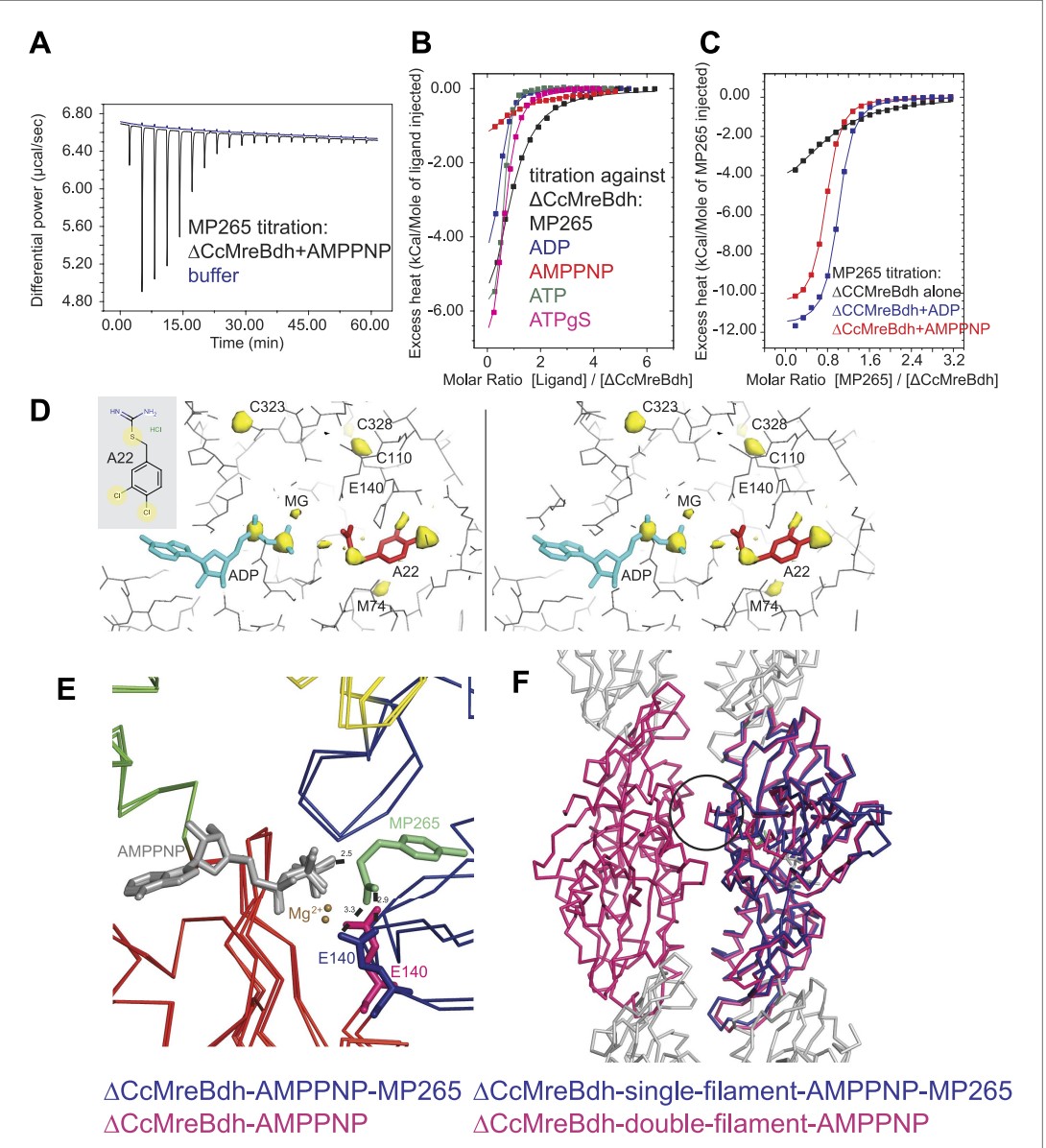

**Figure 7**. Mechanism of the inhibitory action of A22 and MP265 on MreB. (**A**) Representative raw ITC data. Binding of MP265 to ΔCcMreBdh in the presence of 0.5 mM AMPPNP, measured at 25°C (black). Corresponds to red curve in *Figure 7C*. Control heats for the titration of MP265 into dialysis buffer at 25°C are shown in blue. (**B**) ITC measurements and curve fits for different nucleotides and MP265. Black: MP265 ($K_d$ 26 μM, ΔH −7.9 kcal/mol); blue: ADP ($K_d$ 4 μM, ΔH −5.3 kcal/mol); red: AMPPNP ($K_d$ 65 μM, ΔH −4.4 kcal/mol); olive: ATP ($K_d$ 1 μM, ΔH −6.0 kcal/mol); magenta: ATPγS ($K_d$ 4.5 μM, ΔH −7.7 kcal/mol). Binding decreases as follows: ATP > ADP ~ ATPγS >> AMPPNP. Note that MP265 binds 20–30 times tighter in the presence of nucleotide, as shown in *Figure 7C*. (**C**) ITC measurements and curve fits for MP265 binding in the presence of different nucleotides. Measured at 25°C for the binding of MP265 to ΔCcMreBdh alone (black, note separate but identical in design experiment to black curve in **B**) and in the presence of 0.5 mM ADP (blue) or 0.5 mM AMPPNP (red). Solid lines are the fit to data having the following parameters; ΔCcMreBdh and MP265 alone (black) $K_d$ 27 μM, ΔH −6 kcal/mol; ΔCcMreBdh and MP265 + 0.5 mM ADP (blue) $K_d$ 1.3 μM, ΔH −12 kcal/mol; CCM2DM and MP265 + 0.5 mM AMPPNP (red) $K_d$ 1.7 μM, ΔH −11 kcal/mol. Similar data were obtained (not shown) for binding experiments at 15°C where affinities were 19, 0.6 and 0.7 μM respectively. MP265 binds stronger in the presence of di- and tri-phosphate nucleotides than without. (**D**) Stereograph of the active site ADP-bound ΔCcMreBdh in complex with inhibitor A22 (s_2_a), showing the anomalous difference correlates well with the positions of nucleotide (light blue) and inhibitor A22 (red). The structure of A22 is shown in the inset (top left corner). (**E**) Superposition of AMPPNP-bound ΔCcMreBdh crystallised as double protofilaments (d_3) and AMPPNP-bound ΔCcMreBdh in complex with inhibitor MP265 crystallised as a single protofilament (s_3_i). The inhibitor binds to the catalytic residues E140 (blue) and the gamma phosphate thereby inhibiting nucleotide hydrolysis. It also blocks the exit channel for the phosphate. (**F**) Superposition of AMPPNP-bound ΔCcMreBdh crystallised as double protofilaments (d_3, shown in pink) and AMPPNP-bound ΔCcMreBdh in complex with inhibitor MP265 crystallised as a single protofilament (s_3_i, shown in blue). The inhibitor weakens the inter-protofilament interface by displacing the major dimerization helix (indicated by a black circle).

**Table 2.** Crystallographic data

| Name | Tray | Protein | SG | Cell [Å] | Res. [Å] | Source* | $R_{pim}$† | CC1/2‡ | $R/R_{free}$ § | PDB | Model in ASU# | Filament type | |
|---|---|---|---|---|---|---|---|---|---|---|---|---|---|
| **d_e** | 28 | ΔCcMreBh | I2 | 52.3, 98.2, 90.3, β = 98.1° | 2.0 | I04-1 | 0.075(0.236) | 0.858 | 0.20/0.23 | 4cze | 1 MreB, 1 $PO_4$ | double filament, empty | |
| **s_2** | 93 | ΔCcMreBdh | P2$_1$ | 51.5, 71.8, 53.7, β = 100.4° | 1.6¶ | Fr-E | 0.011(0.063) | 0.990 | 0.17/0.20 | 4czf | 1 MreB, 1 MgADP | single filament, ADP | |
| **s_2_a** | 96 | ΔCcMreBdh | P2$_1$ | 50.7, 73.7, 54.2, β = 102.4° | 1.5 | Fr-E | 0.030(0.460) | 0.752 | 0.16/0.22 | 4czg | 1 MreB, 1 MgADP, 1 A22 | single, ADP, A22 | |
| **s_2_i** | 106 | ΔCcMreBdh | P2$_1$ | 50.6, 73.7, 53.7, β = 102.4° | 1.6 | Fr-E | 0.041(0.416) | 0.897 | 0.20/0.23 | 4czh | 1 MreB, 1 MgADP, 1 MP265 | single, ADP, MP265 | |
| **s_e** | 136 | ΔCcMreBdh | P2$_1$ | 52.0, 69.3, 52.2, β = 99.6° | 1.8 | I04 | 0.035(0.210) | 0.884 | 0.18/0.22 | 4czi | 1 MreB | single, empty | |
| **d_3** | 144 | ΔCcMreBdh | P2$_1$2$_1$2$_1$ | 51.5, 101.9, 122.7 | 2.0 | id23eh1 | 0.047(0.253) | 0.847 | 0.18/0.23 | 4czj | 2 MreB, 2 MgAMPPNP | double, AMPPNP | |
| **s_3_i** | 235 | ΔCcMreBdh | P2$_1$2$_1$2$_1$ | 51.7, 73.2, 84.7 | 2.6 | id14eh4 | 0.042(0.256) | 0.818 | 0.20/0.25 | 4czk | 1 MreB, 1 MgAMPPNP, 1 MP265 | single, AMPNP, MP265 | |
| **m_2** | 447 | ΔCcMreBdmh | P3$_2$ | 68.7, 68.7, 68.7 | 1.6 | I03 | 0.033(0.523) | 0.560 | 0.20/0.23 | 4czl | 1 MreB, 1 MgADP | monomeric, ADP | |
| **m_3** | 452 | ΔCcMreBdmh | P4$_3$2$_1$2 | 67.8, 67.8, 320.5 | 2.3 | I02 | 0.019(0.466) | 0.539 | 0.21/0.28 | 4czm | 2 MreB, 2 MgAMPPNP | monomeric, AMPPNP | |

*I04-1, I04, I03 and I02 beamlines at Diamond Light Source, Harwell, UK. Id23 and id14 at ESRF, Grenoble, France. Fr-E refers to in house data collection on a Rigaku Fr-E Superbright + generator equipped with MarDTB imageplate.

†Numbers in parentheses refer to the highest resolution bin.

‡correlation of random half-datasets in the highest resolution bin, as implemented in SCALA.

§5% of reflections were randomly selected for the calculation of the $R_{free}$ value and excluded from all refinement procedures.

#ASU: asymmetric unit of the crystal lattice. Water molecules (not tabulated) were automatically picked with phenix.refine and manually checked.

¶Dataset resolution limited by diffraction geometry, not the crystal.

protofilaments are able to interact with bitopic proteins, such as RodZ (*van den Ent et al., 2010*). In vivo evidence shows that the antiparallel arrangement is not an in vitro artefact and does indeed reflect the filament architecture in *E. coli* (*Figures 4 and 5*). Filament formation could have a synergistic effect on the elongasome by bringing together components that reach a higher local concentration through the interaction with polymeric MreB. Given the almost surprising high degree of similarity between the structures of MreB from mesophilic and thermophilic organisms (*Figure 1A*), it is expected that the antiparallel arrangement of MreB protofilaments is conserved among the bacterial kingdom. This has far reaching consequences for the molecular mechanism of MreB's function. Actin, MamK, ParM, and the other plasmid-segregating, actin-like proteins form pairs of protofilaments with subunits facing the same direction (*Ozyamak et al., 2013b*). In F-actin, the polar ends exhibit different growth and shrinkage rates and selectively interact with binding partners that affect filament protection or nucleation (*Pollard and Mooseker, 1981*; *Wegner, 1982*; *Wang, 1985*; *Pollard, 1986*; *Selve and Wegner, 1986*; *Theriot and Mitchison, 1991*). Pairs of parallel ParM protofilaments extend from the ParRC-interacting end (*Gayathri et al., 2012*). Discrimination of either end has to be absent in the antiparallel arrangement found in the doublets of MreB protofilaments (*Figure 1—figure supplement 2*) and hence MreB filaments must grow and shrink from both ends. The bidirectional movement of MreB filaments in vivo is in agreement with the above hypothesis (*Dominguez-Escobar et al., 2011*; *Garner et al., 2011*; *Reimold et al., 2013*). However, previously reported treadmilling of CcMreB molecules through filaments (*Kim et al., 2006*; *Biteen et al., 2011*) assumes an overall polarity that, as we show, is absent in the filament. It has to be determined whether MreB filaments fluctuate at a steady state, growing and shrinking from both ends, or exhibit dynamic instability leading to catastrophic disassembly, as described in detail for ParM (*Garner et al., 2004*; *Gayathri et al., 2013*).

All MreB proteins studied to date by EM appear in pairs rather than isolated single protofilaments, indicating that double protofilaments could be more stable and/or that filament nucleation requires inter-protofilament contacts. In vitro, single CcMreB protofilaments can only be formed upon disruption of the protofilament interface (*Figure 2D*). Furthermore, the equivalent mutation in vivo renders the protein non-functional (*Figure 4*). In the presence of the antimicrobial agents A22 and MP265, MreB forms single but no double protofilaments (*Figure 7*). By dislocating one of the dimerisation helices and thereby weakening the inter-protofilament interface, the inhibitor might destabilise the MreB filament. This would explain the disruption of MreB filaments by A22 as reported previously (*Bean et al., 2009*; *Dye et al., 2011*).

Association of protofilaments into filaments is unique for each actin-like protein and appropriate for their function. F-actin forms parallel, two-stranded helical filaments with a right-handed twist (*Huxley, 1963*), plasmid segregating protein ParM forms parallel helical filaments with a left-handed twist, that arrange in an antiparallel fashion (*Popp et al., 2008*; *Galkin et al., 2009*; *Gayathri et al., 2012*, *2013*), whereas cell division protein FtsA forms pairs of straight filaments with unknown orientation (*Szwedziak et al., 2012*). Low-resolution images of actin-like protein Alp12 suggest that two pairs of parallel filaments twist around each other in an antiparallel fashion (*Popp et al., 2012*), whereas MamK and Crenactin form a two-stranded, non-staggered helix (*Ozyamak et al., 2013a*; *Izoré et al., 2014*; *Lindås et al., 2014*). Despite these variations in overall filament architecture, the head-to-tail arrangement of subunits in a single protofilament is surprisingly conserved, given the low sequence similarity (~15%). In addition, all protofilaments interact with each other using their flat sides, irrespective of the filament architecture. Moreover, the propeller twist required for filament dynamics in response to nucleotide binding and hydrolysis is preserved. The result of the propeller twist is the closure of domains I and II and the flattening of one side of the molecule that makes the inter-protofilament interface (*Figure 6C*, *Figure 6—figure supplement 2*). The recently published molecular dynamics simulations of MreB (*Colavin et al., 2014*) are in agreement with this observation. Furthermore, detailed structural studies of the different nucleotide states of the best studied actin-like proteins MreB and ParM show that the polymerisation cycle is fairly conserved, with the largest domain movement taking place upon the transition from the monomeric state to the polymeric form (*Figure 6*, [*Gayathri et al., 2012*, *2013*]). In both proteins, the active site geometry in the filament conformation favours nucleotide hydrolysis, which is driven by a catalytic glutamate (E140 in CcMreB and E148 in ParM) that positions the catalytic water in line with the gamma phosphate. The co-crystal structures of AMPPNP-bound CcMreB with the inhibitor confirm the critical role of this catalytic residue E140 (*Figure 7*). In the AMPPNP-bound monomer conformation, no catalytic water is present in CcMreB (*Figure 6F*), whereas the catalytic water in monomeric AMPPNP-bound ParM has moved away from the ideal geometry, thus hampering hydrolysis (*Gayathri et al., 2013*).

It has been reported previously that MreB can polymerise in the presence of either ATP or GTP (*Esue et al., 2006*; *Popp et al., 2010b*; *van den Ent et al., 2001*). From the ΔCcMreBdh co-crystal structures with ADP and AMPPNP, it is clear that the active site could well accommodate a guanidine instead of an adenine as the major contacts are with the ribose and phosphate moieties of the nucleotide, a feature, again, shared with ParM (*Gayathri et al., 2013*).

Taken together, our findings describe a novel architecture of actin-like protofilament pairs that is unprecedented among the actin family of proteins, whether of eukaryotic or prokaryotic origin. The straight, antiparallel arrangement of MreB protofilaments ensures that each subunit can interact with the membrane and membrane-bound components of the elongasome. The non-polar nature of MreB filaments explains common assembly and disassembly kinetics for both filament ends. Future experiments will explore the implications of the antiparallel architecture of MreB on cell wall synthesis and investigate whether MreB dictates bidirectional growth of the cell wall.

## Materials and methods

### Cloning, expression, and purification of MreB proteins from *C. crescentus* and *T. maritima*

Genes of *Caulobacter crescentus* MreB (GenBank: ACL95077.1) and *Thermotoga maritima* MreB1 (GenBank: AAD35673.1) were cloned into the small T7-vector pHis17 (Bruno Miroux, personal communication) by means of PCR and restriction cloning, encoding for the his-tagged proteins: ΔCcMreBh (CCM2), ΔCcMreBdh (pFE397), and ΔCcMreBdmh (CCM3). Full-length, non-tagged CcMreB was

cloned as an N-terminal His-SUMO fusion into pET28a (Kan^R) using In-Fusion technology (Clontech). The vector encoding MGSSHHHHHH-SUMO from pET28a (POPINS, http://www.oppf.rc-harwell.ac.uk/OPPF/protocols/cloning.jsp) was linearised with HindIII and KpnI, purified and mixed with the cloning enhancer treated insert encoding full-length CcMreB for the In-Fusion reaction, resulting in plasmid pFE403. Protofilament interface mutations were introduced by site-directed mutagenesis, essentially as described below for the mutants used in genetic complementation. Full-length non-tagged TmMreB was cloned as a C-terminal Intein–Chitin binding domain fusion into NdeI/SapI linearised pTXB1 (NEB, amp^R) using In-Fusion (Clontech), resulting in plasmid pFE349. The gene encoding for the C-terminal catalytic domain of human SUMO1/sentrin specific peptidase (SENP) was cloned into BamHI/NotI sites of pGex-6p-1 (amp^R) and was a gift from David Kommander (MRC-LMB). All plasmids were verified by sequencing and are listed in *Supplementary file 1*.

## CcMreB

Full-length CcMreB was expressed from pFE403 (Kan^R) and the inter-protofilament mutant V118E in full-length CcMreB was expressed from pFE542 (Kan^R) in C41 cells (Lucigen). Both proteins are fused to an N-terminal His-SUMO tag. Protein expression was induced with 1 mM IPTG and cells were grown in 2xTY medium at 18°C for 8 hr. Bacteria were pelleted by centrifugation, resuspended in buffer A (50 mM Tris–HCl, 300 mM NaCl, 10% glycerol, 5 mM TCEP, pH 8.0), supplemented with DNase (Sigma), RNase (Sigma), and protease inhibitor tablets (Roche) and lysed using a cell disruptor (Constant Systems) at 20 kpsi. The lysate was cleared by centrifugation in a 45Ti rotor (Beckman) at 40k rpm for 25 min and loaded at 2 ml/min on two 5 ml HisTrap columns (GE healthcare). The columns were washed with 5% buffer B (1M imidazole, in buffer A, pH 8.0) and non-tagged CcMreB was eluted from the column after overnight incubation with Sumo protease (GST-SENP). The eluted protein was passed through a GST column to catch away the Sumo protease. The protein was purified from higher order oligomers using size-exclusion (S300HR, GE Healthcare) equilibrated in buffer C (20 mM CHES, 300 mM NaCl, 10% glycerol, 2 mM TCEP, pH 9.5) and concentrated to ~10 mg/ml using Centriprep YM-10 concentrators before being frozen into small aliquots. Electrospray masspec for full-length WT protein gave 36,691 Da (36,682 Da expected).

## ΔCcMreBh

The protein was expressed overnight at 25°C in *E. coli* C41(DE3) cells (Lucigen) with 1 mM IPTG. Cells were resuspended in buffer A (50 mM Tris, 300 mM NaCl, 10% wt/vol glycerol, 5 mM TCEP, pH 8.5) and then opened using a Constant Systems cell disruptor, set at 30k psi pressure. After clearing the lysate in a 45Ti rotor (Beckman) at 35,000 rpm it was loaded at 2 ml/min on two 5 ml HisTrap columns (GE Healthcare). Increasing amounts of buffer B (1 M imidazole, pH 8.0) were applied in steps of 2, 5, 10, 30 and 100%, with most of the protein eluting in pure form at 30% B. The pooled fractions were concentrated to around 2–5 ml with Centriprep YM-10 concentrators (Millipore) and applied to a Sephacryl S300HR column, equilibrated in buffer A. Fractions from the size exclusion column were checked by SDS-PAGE, pooled and re-concentrated to around 20–30 mg/ml using Centriprep YM-10 concentrators. Typically, around 15 mg of final product were obtained from 12 L culture. Electrospray masspec gave 36,916 Da (36,911 Da expected).

## ΔCcMreBdh

Expression and purification followed that of ΔCcMreBh, with the following alterations. Expression was performed at 15°C overnight. Elution from the HisTrap nickel resin peaked at 10% B. The protein elutes with bound ADP from the final S300HR column if no EDTA is used in buffer A. (This was apparent from the 260/280 nm ratio and the crystals obtained from the material, structures: s_2, s_2_a, and s_2_m, see *Table 2*). When 5 mM EDTA was included in buffer A for the Sephacryl S300HR run, the protein eluted without bound nucleotide. The protein becomes more problematic when free of nucleotide and needs to be kept cold in order to avoid precipitation, certainly at higher concentrations. Electrospray masspec gave 36,813 Da (36,809 Da expected).

## ΔCcMreBdmh

Expression and purification followed that of ΔCcMreBh, with the following alterations. Expression was performed at 15°C overnight. Elution from the HisTrap nickel resin peaked at 30% B. When 5 mM EDTA was included in buffer A for the Sephacryl S300HR run, the protein eluted without bound nucleotide. When free of nucleotide, the protein becomes more problematic and needed to be kept cold at all times in order to avoid precipitation, certainly at high concentrations.

## TmMreB

Full-length TmMreB was expressed as an Intein-CBD-tagged protein fusion from plasmid pFE349 and purified as described previously (*Salje et al., 2011*).

## SUMO protease

GST-SENP was expressed in C41 cells (Lucigen) overnight at 15°C. Cells were collected by centrifugation and lysed in buffer A (50 mM Tris, 150 mM NaCl, 1 mM EDTA, 5% glycerol, 2 mM DTT, pH 8.5), supplemented with DNase (Sigma), RNase (Sigma), and protease inhibitor tablets (Roche) using a cell disruptor (Constant Systems, at 20 kpsi). Lysate was cleared by centrifugation for 25 min, at 40k rpm, in a Ti45 rotor (Beckman). The supernatant was bound to 20 ml GST sepharose 4B beads (GE Healthcare) in batch, rotating for 2 hr, at 4°C. The beads were thoroughly washed in buffer A, including a wash in buffer A + 500 mM NaCl and the fusion protein eluted in buffer A supplemented with 10 mM reduced glutathione. After concentrating with a Vivaspin 20 concentrator (30 MWCO), the protein was further purified by size-exclusion chromatography on a Sephacryl S200HR column (GE Healthcare), equilibrated in CcMreB buffer A (50 mM Tris, 300 mM NaCl, 10% glycerol, 5 mM TCEP, pH 8.0). The protein was concentrated to 10–15 mg/ml and frozen in aliquots. Typical yields were ~80 mg of protein from 10 L culture.

## Crystallisation, structure determination and refinement

Initial crystallisation conditions were found using our in house nanolitre crystallisation facility (*Stock et al., 2005*). Final crystals were grown in 96-well MRC crystallisation plates (SWISSCI AG), combining 100 nl of protein solutions at 10–30 mg/ml with 100 nl of reservoir solutions, which are listed in *Supplementary file 2*. Crystals were harvested in loops using a dedicated cryoprotecting solution (*Supplementary file 2*) and snap-frozen in liquid nitrogen. In house X-ray diffraction data were collected on an Fr-E Superbright rotating anode generator (Rigaku), equipped with a MarDTB image plate detector (marresearch GmbH). Synchrotron data were collected on beamlines I02, I03, I04 and I04-1 at Diamond Light Source (Harwell, UK) and beamlines id23eh1 and id14eh4 at ESRF (Grenoble, France), as indicated in *Table 2*. Diffraction data were indexed and integrated with XDS (*Kabsch, 2010*) and scaled and merged with SCALA (*Winn et al., 2011*). All structures were solved by molecular replacement using PHASER (*McCoy et al., 2007*), either with *T. maritima* MreB as the search model (*van den Ent et al., 2001*) or a related structure from within this study. Structures were refined using PHENIX.refine (*Afonine et al., 2012*) using recommended settings and standard restraint libraries, alternating with manual checking and rebuilding in MAIN (*Turk, 2013*). Final models were deposited in the Protein Data Bank (PDB) with codes listed in *Table 2*.

## Preparation of lipid monolayers and negative stain electron microscopy

2D lipid monolayers were prepared following a modified version of the protocol of *Kelly et al. (2008)*. A carbon-coated electron microscopy nickel grid was deposited on a drop of *E. coli* total lipid extract (Avanti Polar Lipids) floating on CcMreB polymerisation buffer (12.5 mM Tris-HCl, 7.5, 2 mM ATP, 4 mM $MgCl_2$) in a teflon well. The grid was removed and after the excess of liquid had been gently blotted away, pre-assembled CcMreB filaments were added for 60 s before staining with 2% uranyl acetate. CcMreB polymerisation (0.25 mg/ml) was carried out in a low salt buffer (12.5 mM Tris–HCl, 7.5, 2 mM ATP, 4 mM $MgCl_2$). Imaging was performed using a 120 kV Tecnai 12 electron microscope (FEI) and the data processed with EMAN2 (*Tang et al., 2007*).

## Preparation of MreB bound to liposomes or lipid tubes for cryo-electron microscopy

Liposomes were prepared from *E. coli* total lipid extract (Avanti Polar Lipids) and lipid tubes were formed by mixing *E. coli* total lipid extract with 30% of D-Galactosyl-β1-1'-N-Nervonoyl-D-erythro-sphingosine (C24:1 β-D Galactosyl-Ceramide, Avanti Polar Lipids). 1 mg/ml of lipid was included in polymerisation buffers for both CcMreB and TmMreB (1 mg/ml); CcMreB polymerization buffer: 12.5 mM Tris-HCl, 7.5, 2 mM ATP, 4 mM $MgCl_2$ and TmMreB polymerisation buffer: 100 mM Tris-HCl, 7.5, 200 mM NaCl, 2 mM ATP, 4 mM $MgCl_2$. MreB-coated lipid tubes samples were applied to a thin continuous carbon film grid (Lacey carbon, Cu, 400 mesh) whereas MreB-liposome samples were applied to holey carbon grids (Quantifoil R2/2, Cu/Rh, 200 mesh) both from Agar Scientific and subsequently frozen in liquid ethane using a FEI Vitrobot. Grids were then transferred to either a FEI Tecnai

G2 Polara or a FEI Titan Krios operated at 300 kV. Single frame images were acquired on the G2 Polara with a back-thinned FEI Falcon II detector. Cryo-electron tomography experiments were carried out on a FEI Titan Krios microscope equipped with a Gatan Quantum GIF and K2 Summit direct electron detector operated using SerialEM (*Mastronarde, 2005*). Tomograms were recorded at a nominal magnification of 26000x (4.5 Å/pixel) with a 3° increment and a tilt range of 60 to −60°, keeping the total electron dose below 120e⁻Å⁻². Tilt series data were reconstructed using the IMOD package (*Kremer et al., 1996*). Two-dimensional cryoEM data were processed using EMAN2 (*Tang et al., 2007*). Subtomogram averaging reconstructions were conducted for 16 tubes from 10 different tomograms independently and ab initio using the AV3 package (*Forster et al., 2005*; *Briggs et al., 2009*; *Bharat et al., 2011*). The overall arrangement of the MreB filaments in all analysed tubes was the same. The subtomogram averaging reconstruction from one representative tube is displayed in *Figure 3G*.

## Genetic complementation

Mutated versions of MreB were examined when expressed from a *lac* promoter on a low copy number plasmid in an MreBCD knock-out strain (FB17, *Bendezú and de Boer, 2008*). In addition, the MreBCD knock-out strain carried plasmid pFB112 (tetR), constitutively expressing transcription factor SdiA and plasmid pFB124 (specR), expressing MreCD at 37°C (*Bendezú and de Boer, 2008*). Mutant versions were created following a modified protocol of site-directed mutagenesis (*Liu and Naismith, 2008*) using desalted, partially overlapping primers. Instead of Pfu polymerase, Q5 High fidelity polymerase (NEB) was used that possesses a higher fidelity and processivity than Pfu polymerase. The mutation was included in the overlapping region of the primer pair that has a Tm of 51–57°C. The Tm for the non-overlapping region was 60–63°C (6–10°C higher than the primer pair overlap). A typical PCR reaction contained 25 ng template plasmid and 0.5 µM of each primer, and cycled 12 x, with an annealing's temperature 3°C above the lowest non-overlapping Tm. The PCR products were treated with DpnI for 2 hr and transformed into chemically competent DH5α cells. Sequencing confirmed a 100% success rate. Mutant MreB was transformed into the strain FB17/pFB112/pFB124 and grown for 6 hr in LB supplemented with 250 µM IPTG, 50 µg/ml ampicillin and 50 µg/ml spectinomycin at 37°C. Cells were stained with FM4-64 and examined with a Zeiss laser-scanning confocal microscope (LSM510) equipped with a 63x oil immersion objective lens. Images were processed using ImageJ.

## In vivo cross-linking

On the basis of the gained structural information a single or pair of cysteine residues was introduced into a cysteine-free form of *E. coli* MreB as the only source of MreB as described above. Over night cultures were diluted 1:100 into 10 ml LB supplemented with 50 µg/ml spectinomycin, 50 µg/ml ampicillin and 250 µM IPTG and grown for 6 hr at 37°C. Cultures were cooled down by adding 30% ice and spun at 1000×*g* for 10 min at 4°C. The pellet washed in ice-cold 10 ml PBSG (PBS, supplemented with 0.1% glycerol), spun at 1000×*g* for 10 min at 4°C. The cell pellet was resuspended in 2 ml PBSG, supplemented with 5 mM EDTA. Half of the resuspended cells were subjected to the thiol-reactive cross-linker bismaleimidoethane (BMOE, Pierce) at a concentration of 100 µM from a 20 mM stock in DMSO, whereas the same amount of DMSO was added to the other half of the cells as a control (*Burmann et al., 2013*). After 10 min on ice, the reaction was quenched with 28 mM β-mercaptoethanol and spun for 10 min at 14 krpm at 4°C (Eppendorf table top centrifuge). The pellets were resuspended in 600 µl B-PER (Thermo Scientific), supplemented with 25 mM DTT, 2.5 mM MgCl₂, 4 mM CaCl, lysozyme, DNAse, RNAse, and incubated at RT for 10 min. The lysed cells were mixed with NuPAGE LDS sample buffer (Life technologies) supplemented with 8 M urea and 5 mM buffered TCEP (pH 7) and heated for 10 min 70°C. Cross-linked products were separated from the monomers on a NuPAGE 4–12% Bis-Tris gel (Life technologies) and detected by Western blot using affinity purified α-MreB antibodies.

## Isothermal titration calorimetry

All ITC measurements were performed at 25°C using an auto-iTC 200 instrument (GE Healthcare) in 50 mM Tris, 300 mM NaCl, 10% vol/vol glycerol, 1 mM TCEP, 5 mM MgCl₂, pH 8.0. Samples were stored by the instrument in 96-well microtitre plates at 5°C prior to loading and performing the ITC titrations. Standard experiments used 19 × 2 µl injections of nucleotide or MP265 into ΔCcMreBdh protein preceded by a single 0.5 µl pre-injection. Heat from the pre-injection was not used during fitting. Data were analysed manually in the Origin software package provided by the manufacturer and fit to a single set of binding sites model. All measurements of binding were corrected using control dilution ITC experiments in which the nucleotide or MP265 was injected into appropriate buffer alone.

The small endothermic heats of each injection in these experiments were fitted to a simple linear function that was subsequently subtracted from the equivalent integrated heats of the experiment when protein was present. The concentration of ΔCcMreBdh in the cell was typically ~70 µM while the concentration of nucleotide or MP265 used in the syringe was ~2 mM.

## Acknowledgements

Cloning and protein purifications were done by FE, TI, and JL, electron microscopy was done by TI with help from TAMB for subtomogram averaging, crystal structure determination was done by JL, ITC was done by CJ, complementation studies and in vivo cross-linking were done by FE. FE prepared the manuscript. We thank David Kommander (MRC-LMB, Cambridge UK) for plasmids pOPINS and SENP1, Linda Amos and Andrzej Szewczak for reading the manuscript and for discussions, beamline scientists at ESRF (beamlines Id14 and Id23) and Diamond Light Source (beamlines I04-1, I04, I03 and I02) and Colin Palmer and Shaoxia Chen (MRC-LMB) for help with electron microscopy. Sacha De Carlo and Sonja Welsch (FEI) supported work on LMB's FEI Krios microscope.

## Additional information

### Funding

| Funder | Grant reference number | Author |
| --- | --- | --- |
| Wellcome Trust | 095514/Z/11/Z | Jan Löwe |
| EMBO Long term Fellowship | ALTF 1379-2011 | Thierry Izoré |
| Federation of European Biochemical Societies | | Tanmay AM Bharat |
| Medical Research Council | U105184326 | Fusinita van den Ent, Christopher M Johnson |

The funders had no role in study design, data collection and interpretation, or the decision to submit the work for publication.

### Author contributions

FE, Cloning, Protein purifications, In vivo cross-linking, Preparation of the manuscript, Conception and design, Acquisition of data, Analysis and interpretation of data, Drafting or revising the article; TI, Protein purifications, Electron microscopy, Acquisition of data, Analysis and interpretation of data; TAMB, Subtomogram averaging, Analysis and interpretation of data; CMJ, ITC, Acquisition of data, Analysis and interpretation of data; JL, Cloning, Protein purifications Crystal structure determination, Conception and design, Acquisition of data, Analysis and interpretation of data

## Additional files

### Supplementary files

• Supplementary file 1. Plasmids used in this study.

• Supplementary file 2. Crystallisation conditions.

### Major datasets

The following datasets were generated:

| Author(s) | Year | Dataset title | Dataset ID and/or URL | Database, license, and accessibility information |
| --- | --- | --- | --- | --- |
| Lowe J, van den Ent F | 2014 | *C. crescentus* MreB, double filament, empty | http://www.rcsb.org/pdb/explore/explore.do?structureId=4cze | Publicly available at RCSB Protein Data Bank. |
| Lowe J, van den Ent F | 2014 | *C. crescentus* MreB, single filament, ADP | http://www.rcsb.org/pdb/explore/explore.do?structureId=4czf | Publicly available at RCSB Protein Data Bank. |

| Lowe J, van den Ent F | 2014 | *C. crescentus* MreB, single filament, ADP, A22 inhibitor | http://www.rcsb.org/pdb/explore/explore.do?structureId=4czg | Publicly available at RCSB Protein Data Bank. |
| Lowe J, van den Ent F | 2014 | *C. crescentus* MreB, single filament, ADP, MP265 inhibitor | http://www.rcsb.org/pdb/explore/explore.do?structureId=4czh | Publicly available at RCSB Protein Data Bank. |
| Lowe J, van den Ent F | 2014 | *C. crescentus* MreB, single filament, empty | http://www.rcsb.org/pdb/explore/explore.do?structureId=4czi | Publicly available at RCSB Protein Data Bank. |
| Lowe J, van den Ent F | 2014 | *C. crescentus* MreB, double filament, AMPPNP | http://www.rcsb.org/pdb/explore/explore.do?structureId=4czj | Publicly available at RCSB Protein Data Bank. |
| Lowe J, van den Ent F | 2014 | *C. crescentus* MreB, single filament, AMPPNP, MP265 inhibitor | http://www.rcsb.org/pdb/explore/explore.do?structureId=4czk | Publicly available at RCSB Protein Data Bank. |
| Lowe J, van den Ent F | 2014 | *C. crescentus* MreB, monomeric, ADP | http://www.rcsb.org/pdb/explore/explore.do?structureId=4czl | Publicly available at RCSB Protein Data Bank. |
| Lowe J, van den Ent F | 2014 | *C. crescentus* MreB, monomeric, AMPPNP | http://www.rcsb.org/pdb/explore/explore.do?structureId=4czm | Publicly available at RCSB Protein Data Bank. |

The following previously published dataset was used:

| Author(s) | Year | Dataset title | Dataset ID and/or URL | Database, license, and accessibility information |
| --- | --- | --- | --- | --- |
| van den Ent F, Amos LA, Lowe J | 2001 | MreB from *Thermotoga Maritima*, AMPPNP | http://www.rcsb.org/pdb/explore/explore.do?structureId=1jcg | Publicly available at RCSB Protein Data Bank. |

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
