## [Decision Letter]

Thank you for sending your work entitled “Bacterial actin MreB forms antiparallel double filaments” for consideration at *eLife*. Your article has been favorably evaluated by a Senior editor and 3 reviewers, one of whom is a member of our Board of Reviewing Editors.

The following reviewers have agreed to reveal their identity: Michael Laub; Ethan Garner.

The Reviewing editor and the other reviewers discussed their comments before we reached this decision, and the Reviewing editor has assembled the following comments to help you prepare a revised submission.

Overall, all three reviewers agreed that this is a very nice paper showing that MreB can form an unprecedented, antiparallel arrangement of protofilaments. The use mutagenesis, crosslinking, crystallography, and EM studies yields a very complete story. The exploration of the nucleotide cycle of MreB and data indicating how the small molecule inhibitors A22 and MP265 affect MreB function were also deemed exciting new findings. There was agreement that the paper represents a major advance and will have broad appeal, and is therefore a good candidate for publication in *eLife*.

However, before accepting the paper, there were several issues raised by all or, in some cases, two of the reviewers that should be addressed in a revised manuscript. To summarize, these issues are: 1) concern about the claims made regarding potential dynamic instability of MreB filaments, 2) concerns about the interpretation of the effects of point mutants. These concerns are described in more detail below and represent an integration and merging of comments from all three reviewers. Some of the concerns raised can be addressed through additional experiments, while others might be resolved through clarification of the text.

Major issues:

1a, from Reviewer 2) The in vivo dynamics of MreB assembly remain unclear; no motor for MreB movement has yet been established. Existing evidence demonstrates that the bidirectional movement of MreB in E. coli is cell-wall synthesis dependent (van Teeffelen et al, PNAS 2011), and recent models suggest that MreB filaments are stable in vivo, and that their movement may be the result of a passive movement on the membrane (Ursell et al, PNAS 2014). Other in vitro work fails to see any turnover of MreB after polymers are sheared (in contrast to actin) (Esue et al, JBC 2012). Given that the authors only give qualitative monomeric structural comparisons with ParM as evidence for the proposal that MreB dynamics are driven by dynamic assembly, I am skeptical of their conclusions that MreB operates through a dynamic instability mechanism.

Similarly, the claim that filaments must grow from both ends due to the symmetric nature of the double-protofilaments crystal structure must be considered outside the context of crystallography. For example, actin has not been crystallized as a filament due to non-integer repeat of filament. Similarly, MreB assembly in the presence (this work; [67]) and absence ([82], many others) of liposomes shows that MreB filaments can take on curved and straight conformations. Indeed, the cryo-EM averaging in this work could only be done on lipid rods that constrained MreB to straight filaments. Crystallographic methods are biased toward solving straight filaments, which leads to a concern and a comment. Firstly, the bending of filaments readily seen in vivo (and presumably required for curvature sensing; see Ursell et al, PNAS 2014) must break the symmetry of the anti-parallel filament. The authors must reconcile this in their discussion of possible MreB dynamics. Secondly, it would be intriguing to hear the authors comment on how monomer conformation (and hence nucleotide hydrolysis) is affected by filament bending.

A recent paper simulating Thermotoga MreB molecular dynamics (Colavin et. al, PNAS 2014) finds a nucleotide and polymerization dependent monomer conformation in quantitative agreement with the authors' structures; namely that the monomer hinges via a “propeller twist” around the nucleotide binding site. Interestingly, that paper also mentions that recent F-actin fibre-diffraction models (Oda et al, Nature 2009 and others) reveal similar propeller twists upon polymerization (which can add to the authors' discussion of ParM polymerization in the Discussion section). This is in conflict with the discussion of MreB being more ParM-like, but could serve as a resolution to the comments above.

It is also unclear in the Discussion section what “strained” MreB conformation refers to in the comparison to ParM?

1b, from Reviewer 3) The most important point that should be addressed in this paper is the authors’ speculation on dynamic instability. The authors offer 2 possibilities for the dynamics of MreB, treadmilling, and dynamic instability. They argue as the filaments are not structurally polar, the filaments are likely to display dynamic instability. This is not an accurate comparison. Structural polarity (or assembly) is different than dynamic instability. One is a spatial property; the other is a kinetic and energetic property.

a) Filaments that are structurally polar need not only display treadmilling: 1) Actin filaments at steady state can be seen to fluctuate rather than treadmill. 2) ParM has been shown to elongate bidirectionally by both the Mullins group [Garner, E. C, Campbell, C. S., R. D. Mullins, Dynamic instability in a DNA-segregating prokaryotic actin homolog, Science 306, 1021-1025 (2004)] and the Lowe group [P. Gayathri et al., A bipolar spindle of antiparallel ParM filaments drives bacterial plasmid segregation, Science 338, 1334-1337 (2012)]. 3) In fact, much work [D. Sept, A. H. Elcock, J. A. McCammon, Computer simulations of actin polymerization can explain the barbed-pointed end asymmetry, J Mol Biol 294, 1181-1189 (1999)], even back to the beginnings of the study of actin (early Marie France papers) have gone into understanding why filaments undergo polar growth.

b) Nonpolar filaments, such as intermediate filaments and CreS also are able to fluctuate at steady state.

c) Third: Dynamic instability is an energetic and kinetic property, that can only arise when here is a substantial energy difference between 2 forms of the polymer, and a stochastic switch between states to yield the fast catastrophes. As of yet, the authors have no kinetic or equilibrium data on MreB polymerization to make this claim.

In this reviewer's opinion, it would behoove the authors to include a third possibility that MreB filaments are fluctuating at steady state, growing and shrinking. While the ADP bound MreB may fall off far faster than the ATP form, this difference may not be fast enough to cause catastrophes.

While it is a sound conclusion that MreB filaments are incapable of displaying treadmilling, there is no evidence they display dynamic instability as of yet. It is far more likely that they simply fluctuate at the ends, with the respective ADP / ATP affinities. It would be far safer for the authors to state all three possibilities, especially as photobleaching in B. Subtilis has shown that MreB filaments are remarkable stable [J. Domínguez-Escobar et al., Processive movement of MreB-associated cell wall biosynthetic complexes in bacteria, Science 333, 225-228 (2011)].

2a) Fully complementing fluorescent fusions of MreB (both wild-type and point-mutants) have proven difficult to construct in the lab, suggesting that many single amino-acid perturbations, be it at the interface or not, can be enough to disrupt proper MreB function.

In the text, the authors present the monomeric crystal structure of intra-protofilament point mutant S283D as evidence the double-protofilament assembly is disrupted. I have two questions with this experiment as it stands. First, given the comment above, the manuscript is missing a convincing control demonstrating that that double-protofilament assembly can still occur when minor mutations are introduced in parts of the protein not essential for polymerization (such as the RodZ binding interface?)

Second, wouldn't the prediction of an inter-protofilament mutation be that MreB would be solved as single protofilaments, instead of as a monomer? The authors should address these questions through control experiments in their revision.

I am similarly confused by the A22 results in the paragraph on lines 328-337 in which they introduce two conflicting observations: that A22 stabilizes single protofilaments in A22 AMPPNP condition, and that this therefore destabilizes filaments in vivo. Why doesn't this perturbation lead to non-polymerized monomer crystals?

Also, how do the authors reconcile the conflicting data that A22 at once stabilizes the protofilament, but destabilizes all MreB puncta in vivo?

2b) The mutation V121E is predicted to eliminate inter-protofilament interactions. But does it also impact individual protofilament formation? It seems like the authors should somehow verify that this substitution specifically impacts inter-filament interaction, as predicted by the crystal structure.

3) The data presented nicely indicate that A22 and MP265 do not prevent nucleotide binding, but instead likely prevent release of phosphate from hydrolyzed ATP. Can this model be directly tested biochemically through detection of free Pi levels?

---

## [Author Response]

*1a, from Reviewer 2) The in vivo dynamics of MreB assembly remain unclear; no motor for MreB movement has yet been established. Existing evidence demonstrates that the bidirectional movement of MreB in E. coli is cell-wall synthesis dependent (van Teeffelen et al, PNAS 2011, and recent models suggest that MreB filaments are stable in vivo, and that their movement may be the result of a passive movement on the membrane (Ursell et al, PNAS 2014). Other in vitro work fails to see any turnover of MreB after polymers are sheared (in contrast to actin) (Esue et al, JBC 2012). Given that the authors only give qualitative monomeric structural comparisons with ParM as evidence for the proposal that MreB dynamics are driven by dynamic assembly, I am skeptical of their conclusions that MreB operates through a dynamic instability mechanism*.

We agree with Reviewer 2 that the in vivo and in vitro dynamics of MreB remain unclear. We only mentioned dynamic instability in the context of what is possible given the newly discovered antiparallel, non-polar filament architecture of MreB. Dynamic instability is a possible mechanism for MreB’s disassembly, if it happens in cells, as it would be compatible with an antiparallel protofilament arrangement, which treadmilling is not. To make this clearer, we have tuned down the discussion on dynamic instability as it is outside the scope of the paper and we provide no evidence for what is happening in terms of disassembly other than stating what is possible in principle. We now briefly mention the consequences of non-polar filament ends on filament dynamics at the end of the first paragraph of the discussion.

*Similarly, the claim that filaments must grow from both ends due to the symmetric nature of the double-protofilaments crystal structure must be considered outside the context of crystallography. For example, actin has not been crystallized as a filament due to non-integer repeat of filament. Similarly, MreB assembly in the presence (this work;*
[67]*) and absence (*[82]*, many others) of liposomes shows that MreB filaments can take on curved and straight conformations. Indeed, the cryo-EM averaging in this work could only be done on lipid rods that constrained MreB to straight filaments. Crystallographic methods are biased toward solving straight filaments, which leads to a concern and a comment. Firstly, the bending of filaments readily seen in vivo (and presumably required for curvature sensing; see Ursell et al, PNAS 2014) must break the symmetry of the anti-parallel filament. The authors must reconcile this in their discussion of possible MreB dynamics. Secondly, it would be intriguing to hear the authors comment on how monomer conformation (and hence nucleotide hydrolysis) is affected by filament bending*.

Firstly, the degree of filament bending due to the curvature of the cell membrane in vivo is tiny at the level of subunit interfaces and much smaller than the bent filaments that have been observed by EM. Secondly, filament bending does not necessarily break the symmetry of the anti-parallel protofilaments as they bend perpendicular to the two-fold symmetry axis. Thirdly, as discussed below, the crystal structures of different nucleotide states show CcMreB in straight conformations, indicating that the nucleotide itself is not the sole determining factor of whether a filament is bent or not. This of course is a hornet’s nest – similar discussions have dominated the tubulin field for many years with the ‘lattice model’ and the nucleotide state models fighting it out.

*A recent paper simulating Thermotoga MreB molecular dynamics (Colavin et. al, PNAS 2014) finds a nucleotide and polymerization dependent monomer conformation in quantitative agreement with the authors' structures; namely that the monomer hinges via a “propeller twist” around the nucleotide binding site. Interestingly, that paper also mentions that recent F-actin fibre-diffraction models (Oda et al, Nature 2009 and others) reveal similar propeller twists upon polymerization (which can add to the authors' discussion of ParM polymerization in the Discussion section). This is in conflict with the discussion of MreB being more ParM-like, but could serve as a resolution to the comments above*.

When discussing the propeller twist in the third paragraph of the discussion, we now refer to the TmMreB molecular dynamics of [10]. Secondly, the discussion proceeds with the comparison of the polymerisation cycles of ParM and MreB – not because they are the most similar actin-like proteins, but because they are the ‘best studied actin-like proteins’ that can provide ‘detailed structural studies of the different nucleotide states’

*It is also unclear in the Discussion section what “strained” MreB conformation refers to in the*
*comparison to ParM?*

In response to Reviewer 3 (point #1b, see below) we have deleted the paragraph on dynamic instability and with that removed the confusion of the “strained” MreB conformation.

*1b, from Reviewer 3) The most important point that should be addressed in this paper is the authors speculation on dynamic instability. The authors offer 2 possibilities for the dynamics of MreB, treadmilling, and dynamic instability. They argue as the filaments are not structurally polar, the filaments are likely to display dynamic instability. This is not an accurate comparison. Structural polarity (or assembly) is different than dynamic instability. One is a spatial property; the other is a kinetic and energetic property*.

*a) Filaments that are structurally polar need not only display treadmilling: 1) Actin filaments at steady state can be seen to fluctuate rather than treadmill. 2) ParM has been shown to elongate bidirectionally by both the Mullins group [Garner, E. C, Campbell, C. S., R. D. Mullins, Dynamic instability in a DNA-segregating prokaryotic actin homolog, Science 306, 1021-1025 (2004)] and the Lowe group [P. Gayathri et al., A bipolar spindle of antiparallel ParM filaments drives bacterial plasmid segregation, Science 338, 1334-1337 (2012)]. 3) In fact, much work [D. Sept, A. H. Elcock, J. A. McCammon, Computer simulations of actin polymerization can explain the barbed-pointed end asymmetry, J Mol Biol 294, 1181-1189 (1999)], even back to the beginnings of the study of actin (early Marie France papers) have gone into understanding why filaments undergo polar growth*.

*b) Nonpolar filaments, such as intermediate filaments and CreS also are able to fluctuate at steady state*.

*c) Third: Dynamic instability is an energetic and kinetic property, that can only arise when here is a substantial energy difference between 2 forms of the polymer, and a stochastic switch between states to yield the fast catastrophes. As of yet, the authors have no kinetic or equilibrium data on MreB polymerization to make this claim*.

*In this reviewer's opinion, it would behoove the authors to include a third possibility that MreB filaments are fluctuating at steady state, growing and shrinking. While the ADP bound MreB may fall off far faster than the ATP form, this difference may not be fast enough to cause catastrophes*.

*While it is a sound conclusion that MreB filaments are incapable of displaying treadmilling, there is no evidence they display dynamic instability as of yet. It is far more likely that they simply fluctuate at the ends, with the respective ADP / ATP affinities. It would be far safer for the authors to state all three possibilities, especially as photobleaching in B. Subtilis has shown that MreB filaments are remarkable stable [J. Domínguez-Escobar et al., Processive movement of MreB-associated cell wall biosynthetic complexes in bacteria, Science 333, 225-228 (2011)]*.

When discussing the possible consequences of the antiparallel arrangement of MreB protofilaments on filament dynamics, we might not have been very clear and must have given the reviewer the wrong impression, for which we apologise. As already mentioned briefly, the only conclusion we wanted to draw from our observations is that the antiparallel arrangement of protofilaments dictates non-polar filaments ends and consequently MreB’s inability to treadmill – a finding the reviewer agrees with and calls it ‘sound’. We fully agree with Reviewer 3 that we know too little to draw any conclusions on the dynamic behaviour of MreB, both in vitro and in vivo at this stage. We have deleted the paragraph on dynamic instability and briefly mention the consequences of non-polar filament ends on filament dynamics, including the possibility of a fluctuating state at the end of the first paragraph of the discussion. We would like to thank the reviewer for such an in depth treatment of the problem – clearly more things to do in the future!

*2a) Fully complementing fluorescent fusions of MreB (both wild-type and point-mutants) have proven difficult to construct in the lab, suggesting that many single amino-acid perturbations, be it at the interface or not, can be enough to disrupt proper MreB function*.

We fully agree with the comment in #2a that any alterations to a filament forming protein like MreB can be misleading. Having said that, fluorescent fusions of MreB used in most of the live-cell imaging papers are a much bigger intrusion and hence more likely to affect MreB’s polymerisation behaviour than the carefully chosen single point mutations we present. But, even some of those point mutations either in- or outside the polymerisation interface can affect MreB’s function, as shown in Figure 5—figure supplement 1.

*In the text, the authors present the monomeric crystal structure of intra-protofilament point mutant S283D as evidence the double-protofilament assembly is disrupted. I have two questions with this experiment as it stands. First, given the comment above, the manuscript is missing a convincing control demonstrating that that double-protofilament assembly can still occur when minor mutations are introduced in parts of the protein not essential for*
*polymerization (such as the RodZ binding interface?)*

Alterations to those parts of the protein that were proven not to be essential for polymerisation are the deletion of the N-terminal amphipathic helix and the double mutation F102S, V103G, shown to form double protofilaments by EM and by crystallography (Table 1 and Table 2, Figure 1, Figure 6—figure supplement 1).

*Second, wouldn't the prediction of an inter-protofilament mutation be that MreB would be solved as single protofilaments, instead of as a monomer? The authors should address these questions through control experiments in their revision*.

Many thanks for this great suggestion! We performed EM studies with the inter-protofilament mutation V118E in full-length CcMreB and indeed it forms single protofilaments, now shown in Figure 2.

*I am similarly confused by the A22 results in the paragraph on lines 328-337 in which they introduce two conflicting observations: that A22 stabilizes single protofilaments in A22 AMPPNP condition, and that this therefore destabilizes filaments in vivo. Why doesn't this perturbation lead to non-polymerized*
*monomer crystals?*

The co-crystals of ΔCcMreBdh with A22 or MP265, show that the main dimerization helix is dislocated, resulting in the occurrence of single protofilaments. This, in combination with the lack of in vivo complementation of the interprotofilament mutation EcMreB-V121E, made us propose that part of the inhibitory function of these antimicrobial agents is interference with double protofilament formation, which is essential for MreB’s function in vivo*.* In the crystals, the single protofilaments are stabilised by crystal contacts. The truth is that this is a speculation, of course, and it is phrased like that in the text.

*Also, how do the authors reconcile the conflicting data that A22 at once stabilizes the protofilament, but destabilizes*
*all MreB puncta in vivo?*

We do not propose that A22 stabilises protofilaments; we found that CcMreB-A22 co-crystals exclusively contain single rather than double protofilaments. We do not know whether single protofilaments can actually exist in the cell, but we do know that they are not functional (inter-protofilament mutation). Secondly, the amount of A22 normally used in vivo is high enough to make MreB precipitate, as it is a very small molecule that probably intercalates many proteins including MreB.

*2b) The mutation V121E is predicted to eliminate inter-protofilament interactions. But does it also impact individual protofilament formation? It seems like the authors should somehow verify that this substitution specifically impacts inter-filament interaction, as predicted by the crystal structure*.

Thank you for raising this clarifying question. As mentioned already, we have now included an EM image of negatively stained single protofilaments obtained with full-length CcMreB-V118E shown in Figure 2 (the equivalent mutation in EcMreB, V121E, does not complement an *mreB* deletion strain, Figure 4).

*3) The data presented nicely indicate that A22 and MP265 do not prevent nucleotide binding, but instead likely prevent release of phosphate from hydrolyzed ATP. Can this model be directly tested biochemically through*
*detection of free Pi levels?*

In our hands, purified TmMreB or CcMreB does not hydrolyse ATP and hence we cannot test the effect of the MreB inhibitors on phosphate release. This is a bit reminiscent of the FtsA situation (which we propose is the ‘analogue’ of MreB in the divisome (76)) and we would propose that MreB (like FtsA) requires an external activator for hydrolysis to become measurable. Of course, currently we cannot formally exclude that there is something else wrong, since it is a negative result. The active site of MreB looks complete, though.